# Audiovisual cues must be predictable and win-paired to drive risky choice

Brett A Hathaway[1,2]*, Dexter R Kim[3†], Salwa BA Malhas[3†], Kelly M Hrelja[1],
Lauren Kerker[3], Tristan J Hynes[1‡], Celyn Harris[3], Angela Langdon[2],
Catharine Winstanley[1,3]*

[1]Graduate Program in Neuroscience, Djavad Mowafaghian Centre for Brain Health,
University of British Columbia, Vancouver, Canada; [2]National Institute of Mental
Health Intramural Research Program, Bethesda, United States; [3]Department of
Psychology, Djavad Mowafaghian Centre for Brain Health, University of British
Columbia, Vancouver, Canada

**\*For correspondence:**
brett.hathaway@nih.gov (BAH);
cwinstanley@psych.ubc.ca (CW)

[†]These authors contributed
equally to this work

**Present address:** [‡]Department
of Experimental Psychology,
University of Cambridge,
Cambridge, United Kingdom

**Competing interest:** The authors
declare that no competing
interests exist.

**Reviewing Editor:** Laura A
Bradfield, The University of
Sydney, Australia

## eLife Assessment

This **important** study provides a nuanced analysis of the impact of cues on cost/benefit decision-making deficits in male rats that could have translational relevance to many addictive disorders. The main findings are that cues paired with rewarded outcomes increase the proportion of risky outcomes, whereas risky choice is reduced when cues are paired with reward loss. The experimental data are **compelling**, whereas the computational analysis based on the optimization of different Q-learning models is **solid**. The findings will be of interest to behavioral neuroscientists and clinicians with an interest in risk, decision making, and gambling disorders.

**Abstract** Risky or maladaptive decision making is thought to be central to the etiology of both drug and gambling addiction. Salient audiovisual cues paired with rewarding outcomes, such as the jackpot sound on a win, can enhance disadvantageous, risky choice in both rats and humans, yet it is unclear which aspects of the cue-reward contingencies drive this effect. Here, we implemented six variants of the rat gambling task (rGT), in which animals can maximize their total sugar pellet profits by avoiding options paired with higher per-trial gains but disproportionately longer and more frequent time-out penalties. When audiovisual cues were delivered concurrently with wins and scaled in salience with reward size, significantly more rats preferred the risky options as compared to the uncued rGT. Similar results were observed when the relationship between reward size and cue complexity was inverted and when cues were delivered concurrently with all outcomes. Conversely, risky choice did not increase when cues occurred randomly on 50% of trials, and decision making actually improved when cues were coincident with losses alone. As such, cues do not increase risky choice by simply elevating arousal or amplifying the difference between wins and losses. It is instead important that the cues are reliably associated with wins; presenting the cues on losing outcomes as well as wins does not diminish their ability to drive risky choice. Computational analyses indicate reductions in the impact of losses on decision making in all rGT variants in which win-paired cues increased risky choice. These results may help us understand how sensory stimulation can increase the addictive nature of gambling and gaming products.

## Introduction

The lights and sounds of a casino are physiologically arousing and increase enjoyment, particularly in those with pathological gambling (*Dixon et al., 2014*; *Loba et al., 2001*; *Spetch et al., 2020*).

**eLife digest** Seafront arcades are popular with people of all ages. For many, the flashing lights and sounds of arcade games add excitement and heighten the thrill of trying their luck on a gambling machine. However, these same stimuli are deliberately designed to encourage excessive gambling. They can also distort our perception of wins and losses, particularly in individuals with gambling addiction.

Previous research has shown that pairing stimuli with wins increases risky choice, but the critical features of these cues remain unclear. Hathaway et al. aimed to identify which specific characteristics of the cue–outcome relationship drive risk preference. Addressing this question helps clarify the mechanisms underlying cue-induced risky choice, with important implications for understanding the addictive nature of slot machines.

To investigate the impact of auditory and visual cues on decision-making, Hathaway et al. exposed rats to a series of behavioral experiments known as the rat Gambling Task. In this task, rats were required to make a choice between four options on each trial, by nose-poking in one of four holes after an initial cue. Two options offered smaller rewards (1–2 sugar pellets) but were relatively safe, with shorter and less frequent time-out penalties following losses. The other two options offered larger rewards (3–4 sucrose pellets) but were riskier, as losses resulted in longer and more frequent time-out penalties.

Most rats learned to favor the low-risk, low-reward options, which maximized total reward earned over time. However, when reward delivery was paired with cues, more rats shifted toward the disadvantageous high-risk options. This suggests that a reliable association between cues and reward delivery is a key driver of risky choice. Cue complexity played only a minor role, and pairing cues with both wins and losses still increased risk preference. In contrast, pairing cues with losses alone reduced risk-taking, while randomly presented cues had no effect. Rather than altering the perceived value of rewards, outcome-paired cues appeared to change the rats' sensitivity to losses.

These findings could ultimately benefit individuals who are vulnerable to, or currently experiencing, gambling disorder – particularly those who frequently use slot machines. By clarifying how sensory cues promote risky decision-making, the research of Hathaway et al. may help inform the design of safer gambling products and guide public health policies aimed at reducing harm. However, before such applications can be realized, findings from animal studies must be replicated and validated in humans, ideally in real-world gambling environments such as casinos.

Indeed, gambling-related cues can cause intense cravings in such individuals, and there is increasing concern over their contribution to addiction (*Limbrick-Oldfield et al., 2017*; *Alter, 2017*). Such cues feature prominently in electronic gaming machines (EGMs), which are specifically designed to encourage excessive gambling (*Griffiths, 1993*). In particular, the salient lights and sounds associated with EGMs are thought to facilitate problematic gambling in susceptible individuals and lead to an increased state of immersion as well as overestimation of the number of wins (*Dixon et al., 2014*; *Alter, 2017*; *Murch et al., 2017*). Deficits in cost/benefit decision making are particularly pronounced in individuals who prefer EGMs over other forms of gambling (*Goudriaan et al., 2005*). Together, this evidence suggests that cue-induced impairments in cost/benefit decision making may be a critical risk factor for the development and maintenance of behavioral addictions such as gambling disorder. However, the impact of salient audiovisual cues on decision making has not been well characterized.

One approach to investigate the influence of outcome-paired cues in cost/benefit decision making utilizes the rat gambling task (rGT), a rodent analog of the human Iowa Gambling Task (IGT, *Zeeb et al., 2009*; *Bechara et al., 1994*). In both tasks, optimal performance is attained by avoiding the two high-risk, high-reward options and instead favoring the low-risk options associated with lower per-trial gains. On the rGT, these low-risk, low-reward options result in less frequent and shorter time-out penalties, and therefore more sucrose pellets may be earned overall. The addition of reward-concurrent audiovisual cues leads to a higher proportion of rats establishing a disadvantageous risky decision-making profile (*Barrus and Winstanley, 2016*). A similar effect of reward-paired cues on risky decision making has been observed in humans (*Cherkasova et al., 2018*). Such cues also appear to

lead to inflexibility in decision-making patterns, as indicated by insensitivity to reinforcer devaluation in the cued but not the uncued rGT (*Hathaway et al., 2021*; *Zeeb and Winstanley, 2013*).

Investigating the learning dynamics of the uncued versus cued rGT using a series of reinforcement learning models revealed that potentiated learning from the cued rewards does not drive risk preference on the cued rGT, as might be expected (*Langdon et al., 2019*). Instead, rats on the cued task were relatively insensitive to the time-out penalties, particularly for the risky options featuring lengthy and more frequent penalties.

Several theories could explain these results. One possibility is that higher levels of arousal resulting from exposure to the cues persist through the time-out penalties on subsequent trials and thereby alter the processing of the punishment signal. Alternatively, reward-paired cues may change the representation of task structure, such that punishments are not correctly integrated into the stored action-outcome contingencies as the rats learn to choose between the options. Salient cues may cause rats to represent winning outcomes as different 'states' to losing outcomes, where learning about one state does not generalize to another (*Niv, 2019*). In that case, time-out penalties would not appropriately devalue the risky options and rats would tend to choose the options offering the highest per-trial reward.

Here, to test these theories and further investigate impairments in risky decision making induced by reward-paired cues, we designed several variants of the rGT to specifically manipulate the complexity and contingency of the outcome-paired cues in the task. In the standard-cued task, the audiovisual cues scale in magnitude and complexity with reward size. To determine whether this scaling is a necessary feature to drive risky choice, we implemented an inverse relationship between cue complexity/magnitude and reward size. We next tested whether cuing all outcomes, ostensibly making trial outcomes more similar and perhaps permitting correct integration into each option's learned value, would similarly impact risky choice as solely cuing the wins. To test whether increased sensory stimulation is sufficient to increase risky choice, cues were played randomly on 50% of trials, regardless of outcome. Lastly, we paired cues with losses instead of wins to investigate whether win-paired cues are necessary to drive risky choice. A reinforcer devaluation procedure was also utilized at the end of training to determine which cue-outcome associations would lead to inflexibility in choice. Using reinforcement learning models of choice behavior, we isolated the effect of these cue manipulations on the profile of trial-by-trial learning from wins and losses in the initial sessions of each task and found that audiovisual cues selectively elevate risky choice when those cues are consistently paired with wins, while selectively cuing losses can rescue decision making from risky preferences that might otherwise develop in this task. Parts of this work were included in a doctoral thesis (*Hathaway, 2023*).

## Results

### Win-paired cues drive preference for risky options

Differences in decision making induced by distinct cue-outcome associations were assessed by training separate cohorts of rats (*n* per task = 28–32) on six variants of the rGT (*Figure 1A and B*). Choice profiles were calculated as the percent choice of the four options on the rGT (optimal: P1, P2; risky: P3, P4) averaged from four sessions at the end of training, once a statistically stable baseline was reached (sessions 35–39,+/-2; see 'Methods'). On average, across the rGT task variants, rats showed a preference for the optimal (i.e., reward-maximizing) option P2. However, there was marked variation between the cohorts in preference for the high-risk options P3 and P4. When comparing P1–P4 choice in an omnibus ANOVA, a significant choice x task interaction was observed ($F(13,430) =$ 2.15, p=0.01; *Figure 2A*; Huynh–Feldt corrected degrees of freedom rounded to the nearest integer). Group comparisons showed differences in P1–P4 choice between task versions that consistently paired cues with wins (standard-cued, reverse-cued, outcome-cued) and those that did not (uncued, random-cued, loss-cued; significant differences from uncued and standard cued tasks indicated on *Figure 2A*; see *Supplementary file 1A* for all comparisons).

As is typical for analysis of data from this task and the IGT, an overall decision score was calculated by subtracting the percent choice of the risky high-reward options from the percent choice of the low-risk/low-reward options ([P1 + P2] – [P3 + P4]), such that lower score indicates greater risky choice. Animals with a decision score above zero were designated as 'optimal', whereas rats with negative decision scores were classified as 'risk-preferring'. Average decision score at the end of training

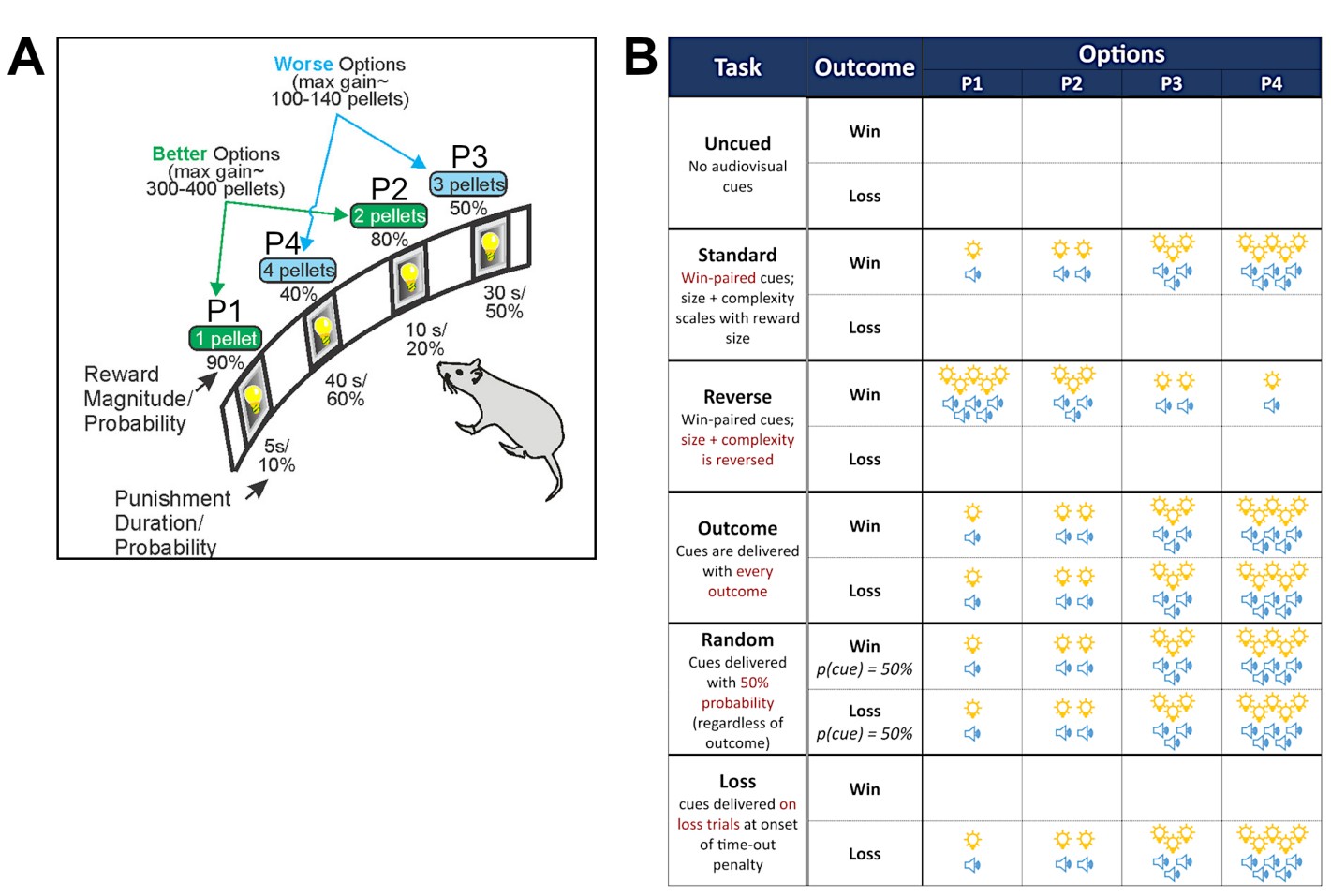

**Figure 1.** The rat gambling task (rGT). (**A**) Schematic of the rGT. A nose poke response in the food tray extinguished the traylight and initiated a new trial. After an inter-trial interval (ITI) of 5 s, four stimulus lights were turned on in holes 1, 2, 4, and 5, each of which was associated with a different number of sugar pellets. The order of the options from left to right was counter-balanced within each cohort to avoid development of a simple side bias (version A [shown]: P1, P4, P3, P2; version B: P4, P1, P3, P2). The animal was required to respond at a hole within 10 s. This response was then rewarded or punished depending on the reinforcement schedule for that option. If the animal lost, the stimulus light in the chosen hole flashed at a frequency of 0.5 Hz for the duration of the time-out penalty, and all other lights were extinguished. The maximum number of pellets available per 30 min session shows that P1 and P2 are more optimal than P3 and P4. The percent choice of the different options is one of the primary dependent variables. A score variable is also calculated, as for the IGT, to determine the overall level of risky choice as follows: [(P1 +P2) – (P3 +P4)]. (**B**) Distinct variants of the rGT. On the uncued variant, no audiovisual cues were present. The standard task featured audiovisual cues that scaled in complexity and magnitude with reward size. The reverse-cued variant inverted this relationship, such that the simplest cue was paired with the largest reward, and vice versa. Audiovisual cues were paired with both wins and losses for the outcome-cued variant. For the random-cued variant, cues were played on 50% of trials, regardless of outcome. Lastly, for the loss-cued variant, cues were only paired with losing outcomes, at the onset of the time-out penalty.

differed significantly between tasks ($F(5, 170)$=6.62, p<0.0001, **Figure 2B** and **Table 1**). In general, the average decision score on tasks featuring win-paired cues was lower than on the uncued task, corresponding to a greater proportion of individual rats with a low or negative decision score, while the random-cued task did not differ significantly from the uncued task. Interestingly, rats trained on the loss-cued task exhibited the greatest preference for the optimal options (highest decision score) among all tasks. Task differences in choice preference may have been driven by increased prevalence of risk-preferring rats on the standard-cued and outcome-cued variants of the rGT, as a significant choice × task × risk status interaction was also observed ($F(13,430)$ = 2.76, p=0.002), and only risk-preferring rats exhibited task differences (risk-preferring: $F(12,96)$ = 1.83, p=0.05; optimal: $F(10,248)$

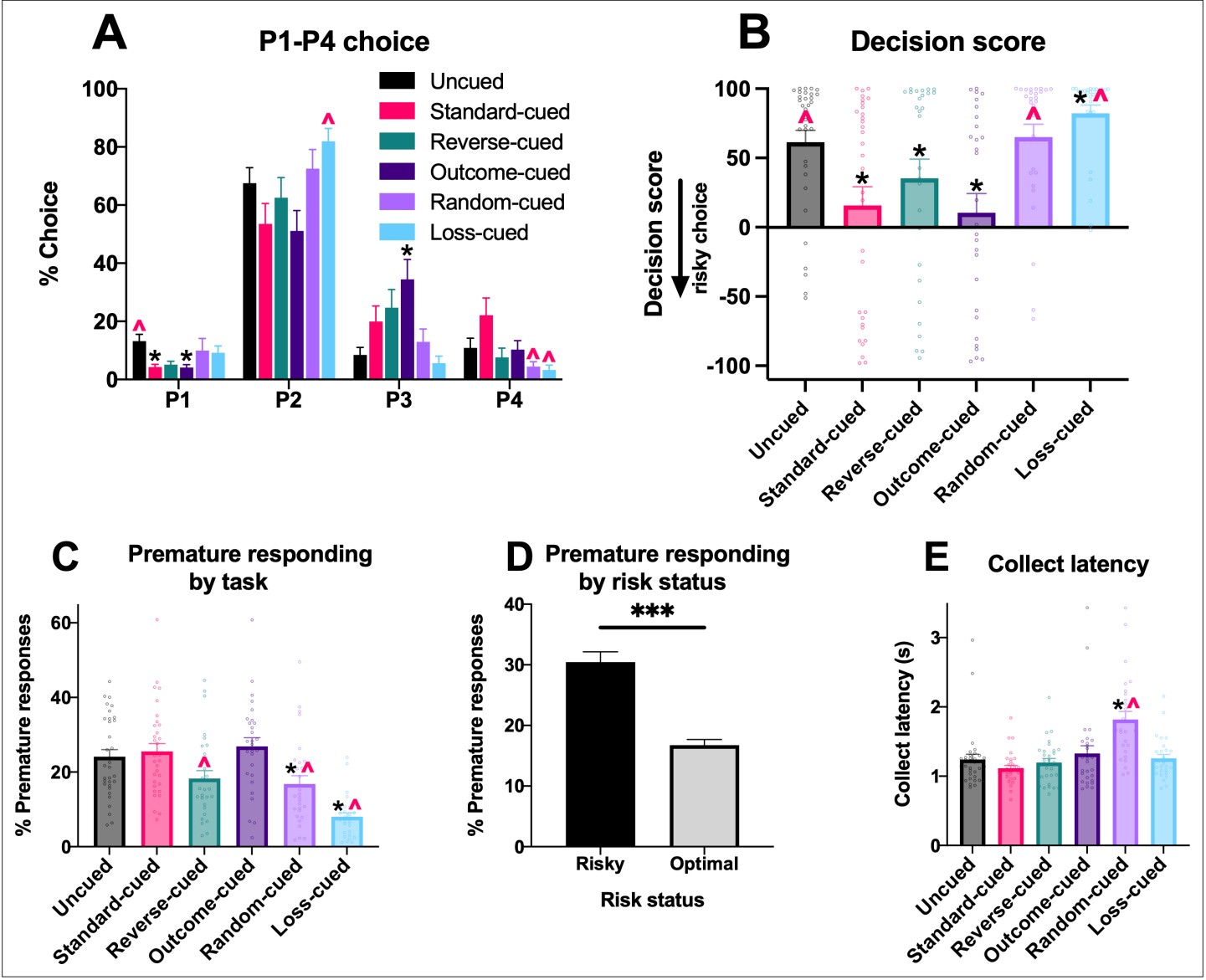

**Figure 2.** Differences in baseline performance between task variants. Comparative baseline performance on variants of the rGT. (**A**) Percent choice of each option in the six rGT task variants. (**B**) Average decision score shows risk preference is significantly modulated by the presence and contingency of outcome-paired cues, with preference for the high-risk options (P3 and P4) strongly enhanced in task variants in which the audiovisual cues scale with outcome magnitude and occur on winning trials. (**C**) Premature responding across the rGT variants, and (**D**) for risky versus optimal decision-makers. (**E**) Latency for reward collection on winning trials across the variants of the rGT. Data are expressed as mean + SEM. Black asterisk indicates significant difference (p<0.05) from uncued task; red caret indicates significant difference from standard cued task. ***p<0.001. N = 176 rats.

The online version of this article includes the following figure supplement(s) for figure 2:

**Figure supplement 1.** Comparative baseline performance for other metrics on variants of the rGT.

---

= 1.12, p=0.35). However, only one post hoc comparison reached marginal significance among risky rats, likely due to the relatively low number of risk-preferring rats for some task variants.

## Premature responding

We next tested whether rats trained on each task variant differed in their level of motor impulsivity. This was measured by the proportion of premature responses made during the 5 s intertrial interval relative to the total number of trials. A significant difference was observed between tasks that was not dependent on risk status ($F_{(5,164)}$ = 5.48, p=0.0001; *Figure 2C*). Post hoc multiple comparisons showed that rats trained on the loss-cued task had the lowest rate of premature responding compared

**Table 1.** Decision score comparisons.

Tukey HSD

| Task comparison | | Mean difference | Significance |
|---|---|---|---|
| Uncued | Standard | **45.51** | <0.0001 |
| | Reverse | **25.32** | 0.006 |
| | Outcome | **51.75** | <0.0001 |
| | Random | –3.29 | 0.99 |
| | Loss | **–22.23** | 0.03 |
| Standard | Reverse | –20.19 | 0.06 |
| | Outcome | 6.24 | 0.95 |
| | Random | **–48.81** | <0.0001 |
| | Loss | **–67.75** | <0.0001 |
| Reverse | Outcome | **26.43** | 0.006 |
| | Random | **–28.62** | 0.002 |
| | Loss | **–47.56** | <0.0001 |
| Outcome | Random | **–55.05** | <0.0001 |
| | Loss | **–73.99** | <0.0001 |
| Random | Loss | –18.94 | 0.11 |

Comparisons of decision score between task variants using Tukey's HSD test. Bolded values indicate a significant difference.

to all other tasks (*Supplementary file 1B*). Rats trained on the reverse-cued and random-cued task variants also exhibited a lower level of premature responding compared to the uncued, standard-cued, and outcome-cued rats. Across all task groups, risk-preferring rats had a significantly higher proportion of premature responses than optimal rats ($F(1,164)$ = 23.41, p<0.0001; *Figure 2D*).

## Other variables

Rats differed in their latency to collect reward across the task variants ($F(5,164)$ = 2.47, p=0.04; *Figure 2E*). Results from the post hoc multiple comparisons are displayed in *Supplementary file 1C*, showing that rats trained on the random-cued task were significantly slower to collect reward than all other rats.

No differences between task variants were observed in latency to choose an option, trials completed, or omissions (all p>0.10; *Figure 2—figure supplement 1*). Across all tasks, risk-preferring rats completed significantly fewer trials than optimal rats ($F(1,164)$ = 99.28, p<0.0001), as expected given that they experienced a higher number of lengthy time-out penalties.

**Table 2.** Devaluation in risk-preferring rats: P1–P4 choice.

| Task | F value | Degrees of freedom | p value |
|---|---|---|---|
| Uncued/random/loss (*n*=7) | **4.17** | 3,18 | 0.04 |
| Standard (*n*=7) | 0.14 | 3,15 | 0.93 |
| Reverse (*n*=3) | 2.98 | 3,6 | 0.12 |
| Outcome (*n*=14) | 1.47 | 3,39 | 0.24 |

Choice × devaluation interactions for each task in risk-preferring rats. Bolded values indicate a significant difference.

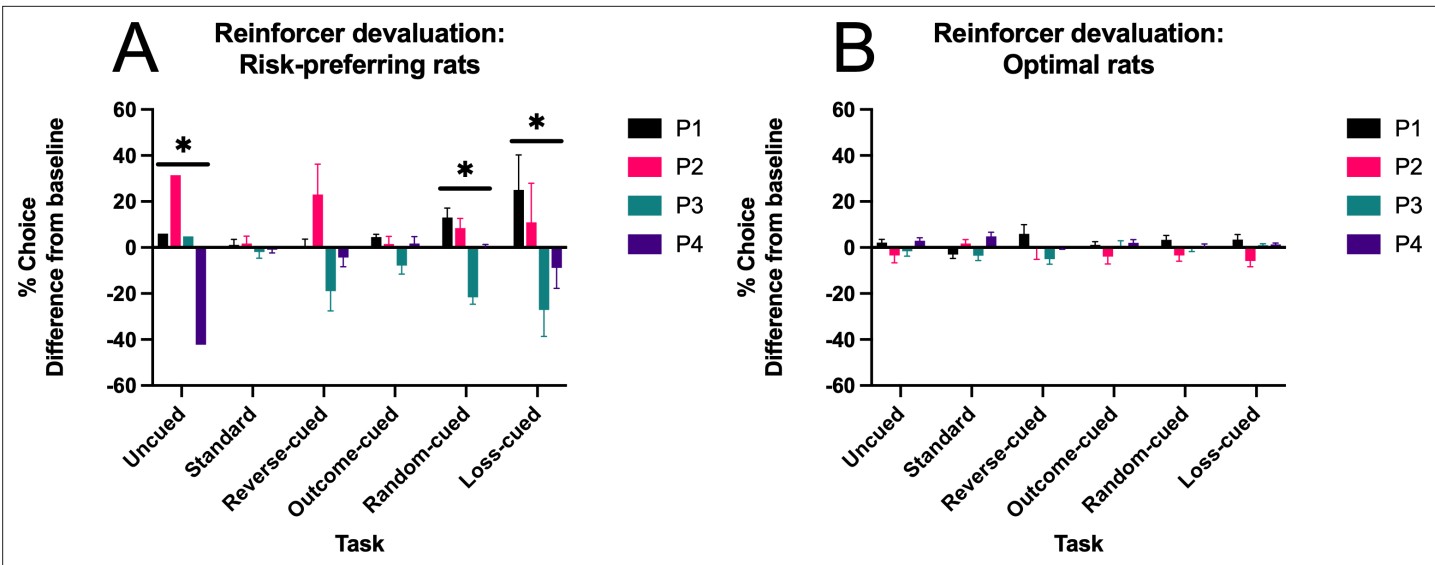

**Figure 3.** Effects of sucrose pellet devaluation on choice preference. (**A**) P1–P4 choice preference after reinforcer devaluation compared to baseline preference for risk-preferring rats. Devaluation did not shift choice patterns selectively in task variants featuring consistent win-paired cues (standard, outcome-cued, reverse-cued). (**B**) P1–P4 choice preference after reinforcer devaluation compared to baseline preference in optimal rats. Reinforcer devaluation induced a slight shift in choice preference, with no differences found between tasks. Data are expressed as the mean change in % choice from baseline + SEM to highlight effects independent of differences in preference for each option between cohorts. Asterisk indicates significant choice × devaluation effect, p<.05. n = 31 risk-preferring rats, n = 95 optimal rats.

## Win-paired cues induce insensitivity to reinforcer devaluation
### Choice

We next tested whether pairing salient audiovisual cues with outcomes on the rGT impacts flexibility in decision making when outcome values are updated. Reinforcer devaluation, in which subjects are sated on the sugar pellet reinforcer prior to task performance (presumably devaluing the outcome), is a common test of flexibility of decision making (*Adams and Dickinson, 1981*). We have previously employed this method to demonstrate that rats trained on the standard-cued task are insensitive to reinforcer devaluation (i.e., choice patterns do not shift despite devaluation of the sugar pellet reward; *Hathaway et al., 2021*). To determine which task variants resulted in inflexible choice patterns, a subset of rats (*N*=126) was subjected to the reinforcer devaluation test in which they received ad libitum access to sucrose pellets for 1 h prior to task performance. Data from the devaluation test were then compared to a separate baseline session during which no experimental manipulation occurred. A significant devaluation × choice × task effect was observed that was dependent on risk status (devaluation × choice × task × risk status: *F*(14, 314)=0.44, p=0.002). This effect was marginally significant in risk-preferring rats (F(13,66) = 1.79, p=0.06). Effects broken down by task for risk-preferring animals can be found in *Table 2*; risk-preferring rats on the uncued, random-cued, and loss-cued tasks were grouped together due to low *n* (1–3 per task).

**Table 3.** Devaluation in risk-preferring rats: Decision score.

| Task | F value | Degrees of freedom | p value |
|---|---|---|---|
| Uncued/random/loss (*n*=7) | **55.49** | 1,6 | 0.005 |
| Standard (*n*=7) | 0.02 | 1,6 | 0.89 |
| Reverse (*n*=3) | 4.88 | 1,2 | 0.16 |
| Outcome (*n*=14) | 3.45 | 1,13 | 0.09 |

Decision score × devaluation interactions for each task in risk-preferring rats. Bolded values indicate a significant difference.

The impact of reinforcer devaluation on choice preference is depicted in *Figure 3A* as a difference in % choice between baseline and devaluation sessions (baseline subtracted from devaluation) for each task variant. This was done to highlight shifts in choice separate from overall group differences in the selection of the different options. Among the risky rats, only those trained on tasks without win-paired cues exhibited changes in choice patterns following reinforcer devaluation. This indicates that pairing audiovisual cues with reward induces some degree of inflexibility in risk-preferring rats. Importantly, pairing cues with losses alone does not elicit rigidity in choice. Thus, in keeping with the observed effect on overall choice patterns, pairing cues with wins has a unique impact on sensitivity to reinforcer devaluation. Although not statistically significant, visual inspection of the reverse-cued task suggests that some choice flexibility may be present, and the study may be underpowered to detect this effect. Nonetheless, win-paired cues that scale with reward size reduce flexibility in choice patterns following reinforcer devaluation.

While the effect of devaluation did not differ by task in optimal rats ($F_{(13,226)} = 0.95$, p=0.50), a marginally significant choice × devaluation effect was observed in these rats ($F_{(3,226)} = 2.62$, p=0.06; see *Figure 3B*), indicating that some degree of shifting occurred in optimal rats that was not influenced by the presence or absence of cues.

The observed shifts in P1–P4 choice resulted in a significant task-dependent shift in decision score in risk-preferring rats (devaluation × task: $F_{(5,25)} = 3.90$, p=0.01) but not optimal rats ($F_{(5,89)} = 1.53$, p=0.19). Results are summarized by task in *Table 3*. Similar to the choice results, only rats trained on tasks without win-paired cues exhibited shifts in risk preference following reinforcer devaluation.

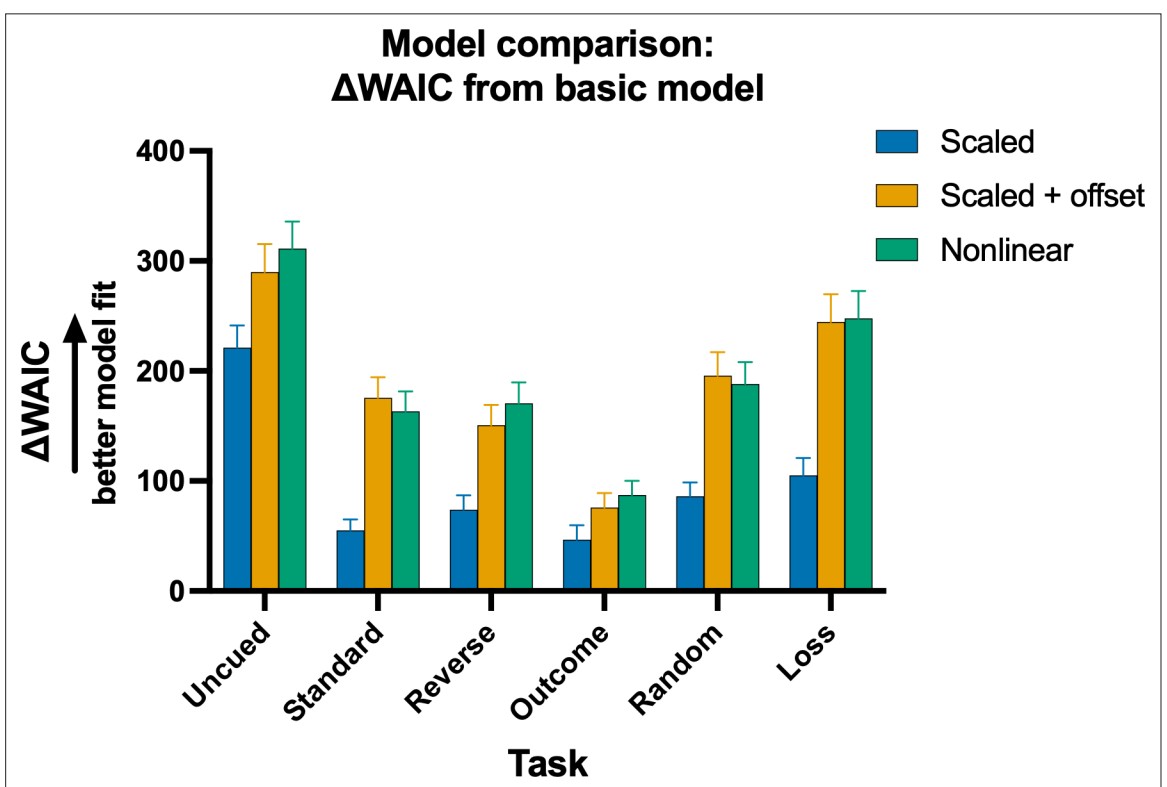

**Figure 4.** Difference in WAIC between each model and the basic model across the rGT task variants. Differences were calculated by subtracting each model's WAIC from the basic model WAIC for that task variant. Larger differences indicate a better explanation of the data. Error bars are SEM of the WAIC difference.

The online version of this article includes the following figure supplement(s) for figure 4:

**Figure supplement 1.** Visualization of how different values of the *m*, *r*, and *b* parameters impact the transformed time-out penalty durations into equivalent cost in pellets.

## Other variables

Latency to collect reward did not shift in response to devaluation ($F(1,114) = 0.55$, p=0.46). Latency to choose an option significantly increased across all tasks ($F(1,114) = 71.38$, p<0.0001), as did omissions ($F(1,114) = 9.75$, p=0.002). Trials decreased in all rats, particularly optimal decision-makers (devaluation × risk status: $F(1,114) = 6.72$, p=0.01; optimal: $F(1,89) = 80.66$, p<0.0001; risky: $F(1,25) = 9.41$, p=0.006). Premature responding significantly decreased across all groups ($F(1,114) = 63.32$, p<0.0001).

## Nonlinear transform best describes the impact of time-out penalties on choice for most rGT variants

These results indicate that salient audiovisual cues reliably paired with wins increase the number of individual rats that display a preference for risky options, and that this pattern of responding is insensitive to devaluation selectively in this subset of rats. Next, we asked how choice preferences on each rGT task variant related to learning dynamics during the initial sessions of each task, and how this differed between risk-preferring and optimal rats.

We investigated differences in the acquisition of each task variant by fitting several reinforcement learning (RL) models to early sessions. Our modeling approach closely follows methods outlined in *Langdon et al., 2019*, in which a much larger dataset (>100 rats per task) was used to develop the RL models applied here. Due to the comparatively small *n* per group in the current study, we limited our model selection to those previously validated in *Langdon et al., 2019*, with minor extensions. As in previous work, models were fit to valid choices from the first five sessions. As training continues, the impact of outcomes on subsequent choice should decline, and parameter values may evolve over time (e.g., decreasing learning rate). To target the period of learning during which outcomes have maximal influence over choice, and parameters likely have fixed values, we limited our analyses to the first five sessions.

To quantitatively examine choice variability during training, we binned sessions 1–5 and 6–10 and analyzed variability in choice patterns across task variants. Analysis of the first five sessions of training revealed a significant shift in decision score across sessions ($F(3, 502)=31.23$, p<0.0001), which differed between task variants (session × task: $F(16, 502)=2.13$, p=0.007). Conversely, while significant differences in overall score were observed between task variants in sessions 6–10 (task: $F(5, 156)=6.81$, p<0.0001), there was no significant variability across sessions (session: $F(3, 481)=2.06$, p=0.10, task × session: $F(15, 481)=0.78$, p=0.71). This indicates that the variability in choice preference (and presumably, learning about outcomes) is maximized in the first five sessions, and there are no obvious differences in the rate of development of stable choice patterns between task variants.

Each of the RL models assumes that choice on every trial probabilistically follows latent $Q$-values for each option, which are updated iteratively according to the experienced outcomes. Winning outcomes ($R_{tr}$) increase $Q$-values in a stepwise manner governed by the reward learning rate ($\eta^{+}$), according to a delta-rule update:

$$Q_x^{new} = Q_x^{old} + \eta^{+} \left( R_{tr} - Q_x^{old} \right)$$

The most basic model uses the same delta-rule update for losing trials, with a separate punishment learning rate ($\eta^{-}$):

$$Q_x^{new} = Q_x^{old} + \eta^{-} \left( 0 - Q_x^{old} \right)$$

In this model, time-out penalties do not explicitly influence the trial-wise decrease in $Q$-values on loss trials. Thus, three different models were designed to determine how time-out penalty duration impacted $Q$-values. Each model tests a different hypothesis as to how time-out penalties ($T_{tr}$) are transformed into an equivalent 'cost' in sucrose pellets.

| Model name | Parameters | Punishment update |
|---|---|---|
| Scaled cost | 4 | $Q_x^{new} = Q_x^{old} + \eta^- \left( -mT_{tr} - Q_x^{old} \right)$ |
| Scaled + offset cost | 5 | $Q_x^{new} = Q_x^{old} + \eta^- \left( b - mT_{tr} - Q_x^{old} \right)$ |
| Nonlinear cost | 5 | $Q_x^{new} = Q_x^{old} + \eta^- \left( b - T_{tr}^r - Q_x^{old} \right)$ |

The scaled cost model assumes a linear relationship between the experienced time-out penalty durations, controlled by parameter $m$. The scaled + offset cost model features an additional offset parameter $b$, allowing for a global increase or decrease in the impact of time-out penalties. Lastly, the nonlinear cost model uses a power law to enable a nonlinear relationship between time-out duration and its relative impact on $Q$-values; the $r$ parameter determines the curvature of this nonlinear transform. *Figure 4—figure supplement 1* illustrates how differing values for the $m$, $r$, and $b$ parameters impact the transformation of the time-out penalty durations. For all three models, a punishment learning rate ($\eta^-$) was estimated to determine the stepwise decrease to $Q$-values following a time-out penalty. Choice probability was determined according to a softmax rule, where the β parameter controls how closely rats' choices follow their latent $Q$-values (lower β value indicates more random choice across the four options; see 'Methods'). Individual subject- and group-level parameters for each model were estimated by hierarchically sampling their posterior distributions for each of the RL models using Hamiltonian MCMC as implemented in Stan (*Carpenter et al., 2017*).

For each task variant, the best-fitting model was assessed using the Watanabe–Akaike information criterion (WAIC; *Watanabe, 2010*). This term assesses model fit whilst also penalizing for model complexity, with lower WAIC indicating a better explanation of the data. Model comparisons were made relative to the basic delta-rule model featuring separate learning rates for wins and losses but no time-out penalty transformation (*Figure 4*). Among the models tested, the nonlinear cost RL model best captured the pattern of choice during the learning phase for four of the six task variants (uncued, reverse, outcome, and loss), with the scaled + offset cost model performing better for the standard- and random-cued task variants. Together, this suggests that for the majority of rats, the subjective cost associated with a loss for each option was not related to time-out penalty duration in a linear manner, at least during the initial sessions of learning.

To confirm that the best-fitting models captured the dominant features of the behavioral data, we simulated the probability of each option on each trial for 40 sessions, using the subject-level model parameter estimates. We then calculated the decision score for sessions 36–40 of the simulated data for the nonlinear model (*Figure 5A*) and the scaled + offset model (*Figure 5B*). While fewer significant differences were observed in the simulated data compared to the actual data (statistical tests available in *Supplementary file 1D and E*), the overall pattern of results was largely preserved, with simulated choices on task variants featuring win-paired cues exhibiting lower decision scores than those from the uncued, random-cued, and loss-cued rGT. Simulated P1–P4 choices are shown in *Figure 5C and D* (see *Supplementary file 1F and G* for statistical tests). For the uncued and loss task variants, differences in choice appeared in P1 rather than the P2 option as in the real data. This may indicate that for some tasks variants, these models were not able to appropriately recapitulate choice preferences between the two optimal options. One potential explanation is that the models were fit to early acquisition sessions, during which P1 sampling is higher. Alternatively, additional processes not captured by the present RL formulations may suppress P1 choice as training continues. Nevertheless, the similarity in differences in decision score demonstrates that the models capture the main finding of cue-induced increase in risk preference. Consistent with the WAIC results indicating that the nonlinear model is the winning model for the majority of the rats, mean differences were larger and more statistical differences were found for the nonlinear model compared to the scaled + offset model.

## Outcome-paired cues differentially impact the learning rate for wins versus losses

Next, we asked how the model parameters that control learning differed between the rGT task cohorts in these early sessions of training. Since both the nonlinear cost and the scaled + offset cost models performed the best for some of the task variants, we compared the group-level mean posterior

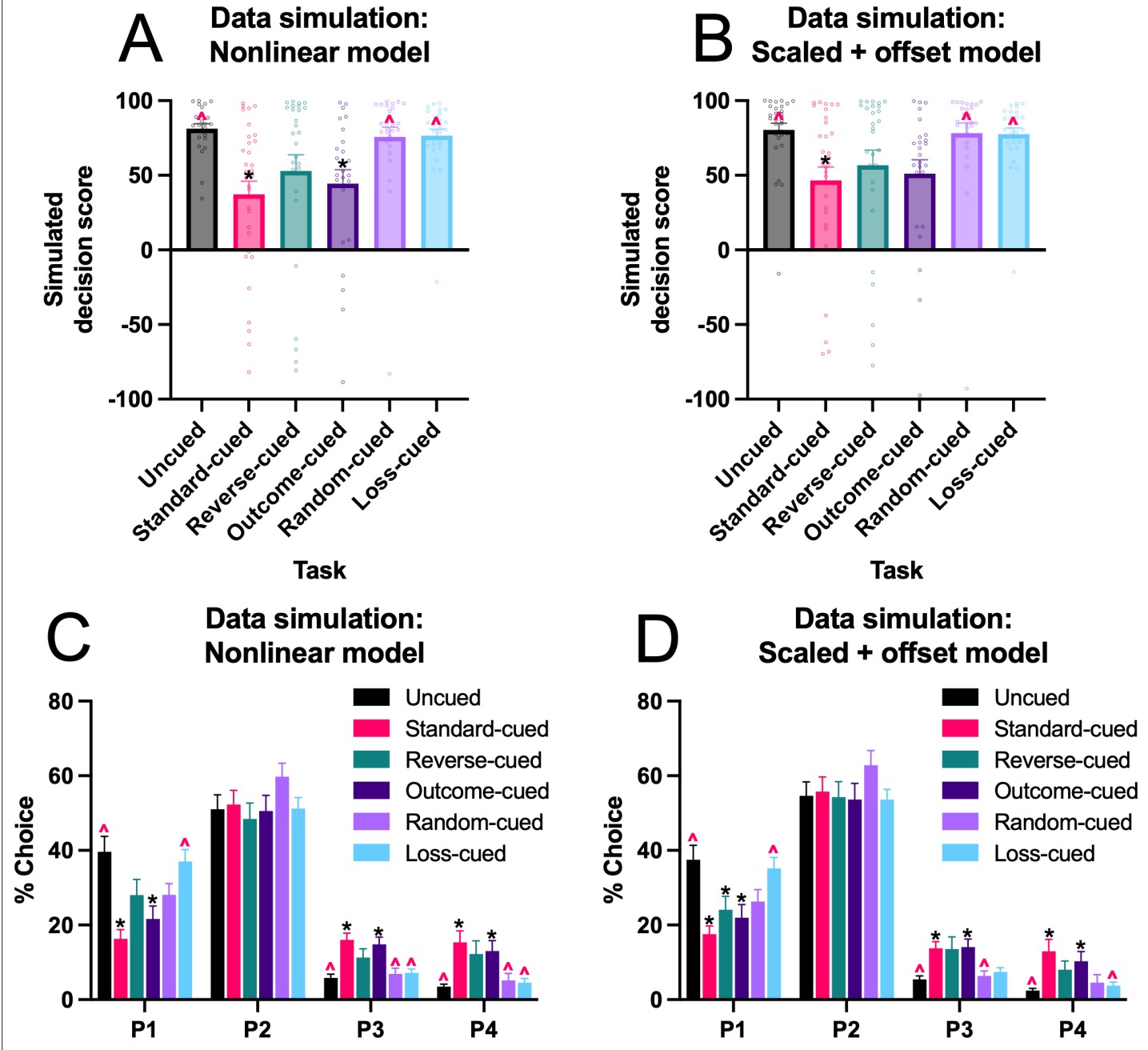

**Figure 5.** Average simulated decision score and P1–P4 choice (sessions 36–40) for the nonlinear and scaled + offset models, simulated with the subject-level parameter estimates for each task variant. (**A**) Average simulated decision score for the nonlinear model. (**B**) As in (**A**), for the scaled + offset model. (**C**) Average simulated P1-P4 choice for the nonlinear model. (**D**) As in (**C**), for the scaled + offset model. Black asterisk indicates significant difference from uncued task; red caret indicates significant difference from standard cued task. N = 165 rats (simulated from subject-level parameter estimates).

estimates for each parameter from both models. Differences between parameter estimates were considered credible when the 95% highest-density interval (HDI) for the sample difference between two mean estimates did not include zero.

Differences between task variants were found in the beta and learning rate estimates for both the nonlinear cost (*Figure 6*) and scaled + offset cost (*Figure 7*) models. All tasks featuring reward-paired cues (standard-, outcome-, reverse-cued tasks) exhibited a lower punishment learning rate than the loss-cued task variant. The punishment learning rate estimates for the uncued and random-cued variants fell between the estimates for the loss-cued task and the variants featuring win-paired cues. Thus,

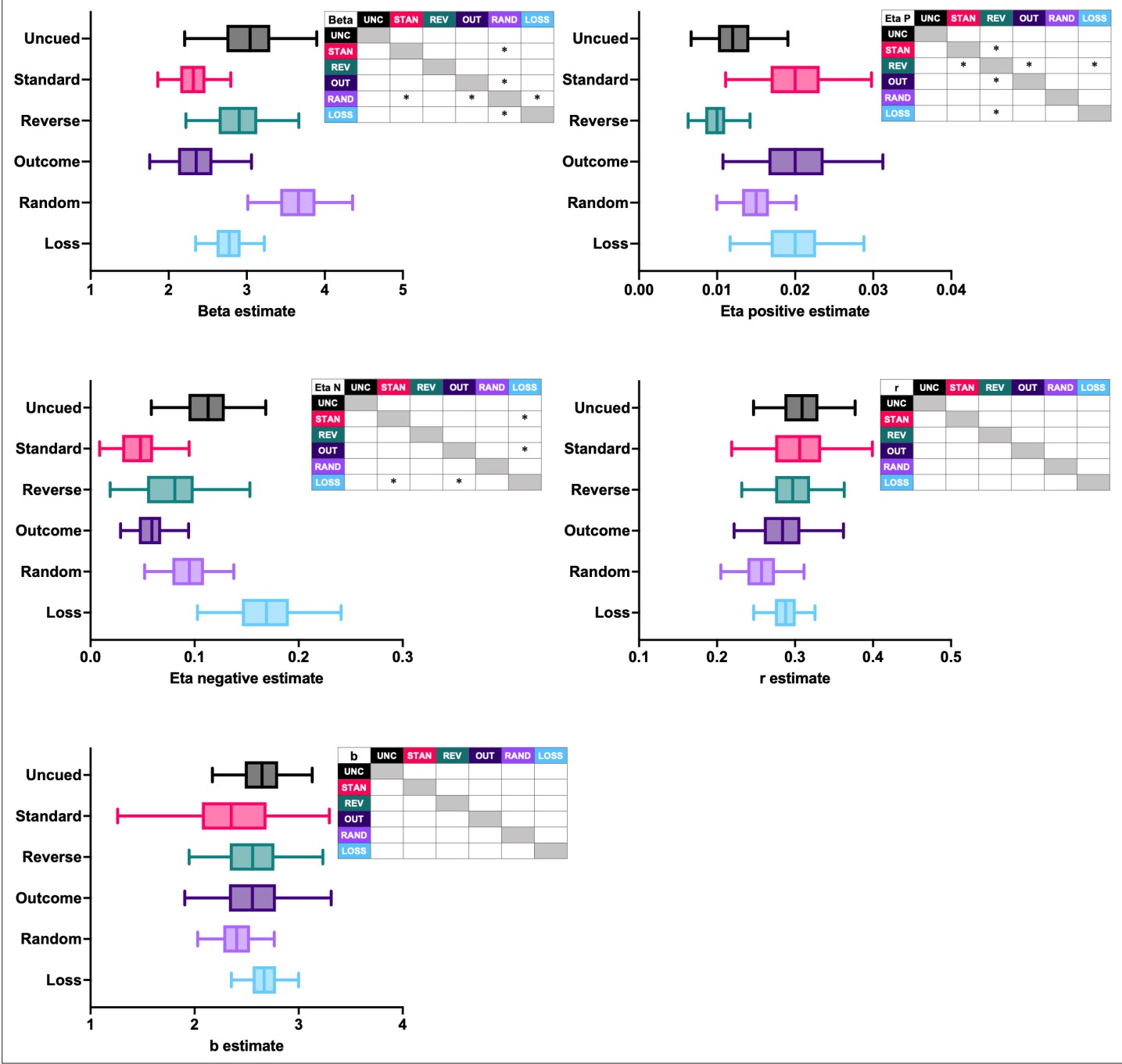

**Figure 6.** Group-level posterior estimates of nonlinear cost model parameters. Asterisks within the inset tables mark parameters for which the 95% HDI of the sample difference did not contain zero, indicating a credible difference. For each distribution, the line demarcates the mean, the box demarcates the interquartile interval, and the whiskers demarcate the 95% HDI. N = 165 rats.

the rank order of the punishment learning rate estimates exactly mirrored the pattern observed in the rats' decision scores at the end of training. This suggests that win-paired cues reduce the impact of losing outcomes to drive risky choice. Given that across both models, the loss-cued task exhibited the highest punishment learning rate, cuing losses may increase their impact on subsequent choice and thereby reduce risky choice.

Learning from wins was also affected by outcome-associated cues. Across both models, the reward learning rate estimate for the reverse-cued task was lower than the other variants featuring predictable win-paired cues; it was also lower than the loss-cued variant in the nonlinear model. These results

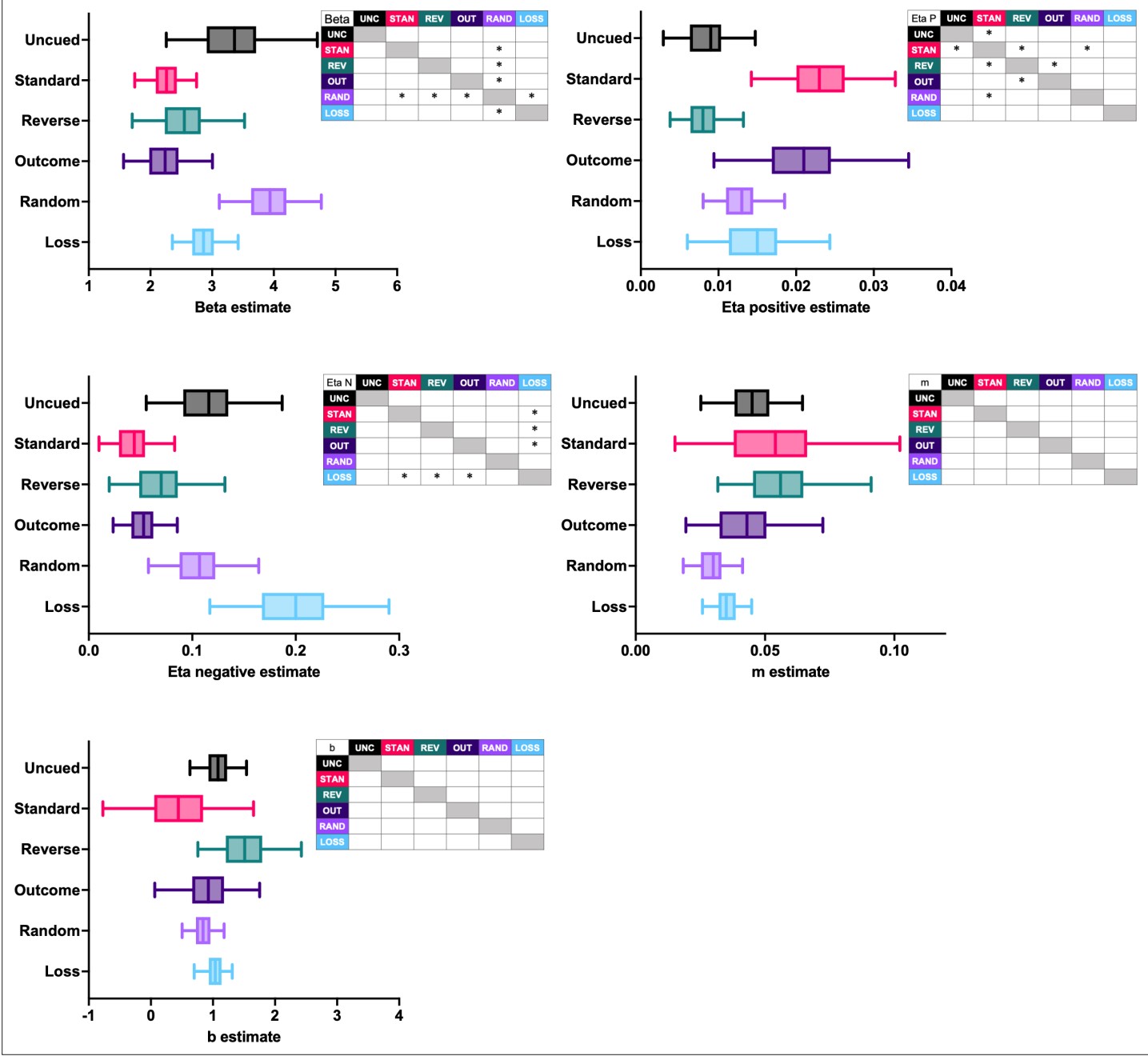

**Figure 7.** Group-level posterior estimates of scaled +offset model parameters. Asterisks within the inset tables mark parameters for which the 95% HDI of the sample difference did not contain zero, indicating a credible difference. For each distribution, the line demarcates the mean, the box demarcates the interquartile interval, and the whiskers demarcate the 95% HDI. N = 165 rats.

The online version of this article includes the following figure supplement(s) for figure 7:

**Figure supplement 1.** Group-level posterior estimates of scaled model parameters.

**Figure supplement 2.** Group-level posterior estimates of scaled reward model parameters.

suggest that the inverted relationship between cue complexity and reward size for the reverse-cued task may have diminished learning from wins.

Generally speaking, the tasks featuring win-paired cues exhibited a lower beta estimate, although a credible difference was only found when compared to the random-cued variant. This may indicate that when win-paired cues are present, choice patterns did not follow latent Q-values as closely.

While we restricted our model selection to those previously validated on larger datasets, the specificity of the main finding to the punishment learning rate may be due to the greater flexibility afforded to loss scaling, rather than a true asymmetry in learning. To test this hypothesis, we fit a model featuring a scaling parameter for rewards, in addition to scaled costs:

$$Q_x^{new} = Q_x^{old} + \eta^+ \left( mRew \times R_{tr} - Q_x^{old} \right),$$

where *mRew* is a linear scaling parameter for reward size. A separate scaling parameter was used for time-out penalty duration (i.e., same as scaled cost model). Group-level parameter estimates (*Figure 7—figure supplement 1*) reflected similar differences in the punishment learning rate and reward learning rate as the scaled cost model (*Figure 7—figure supplement 2*). Furthermore, all 95% HDIs for the *mRew* scaling parameter included 1, indicating that at least at the group level, scaling of reward size across the P1–P4 options closely follows the actual number of earned sucrose pellets. Thus, we find no evidence that our results can be simply attributed to the increased parameterization of losing outcomes.

## Parameters predicting risk preference on the rGT

We next tested whether any of the subject-level parameter estimates in the nonlinear or scaled + offset model could reliably predict risk preference scores at the end of training. In *Langdon et al., 2019*, analyses were conducted to test whether parameters controlling sensitivity to punishment predicted final decision score at the end of training in the uncued and standard cued task variants.

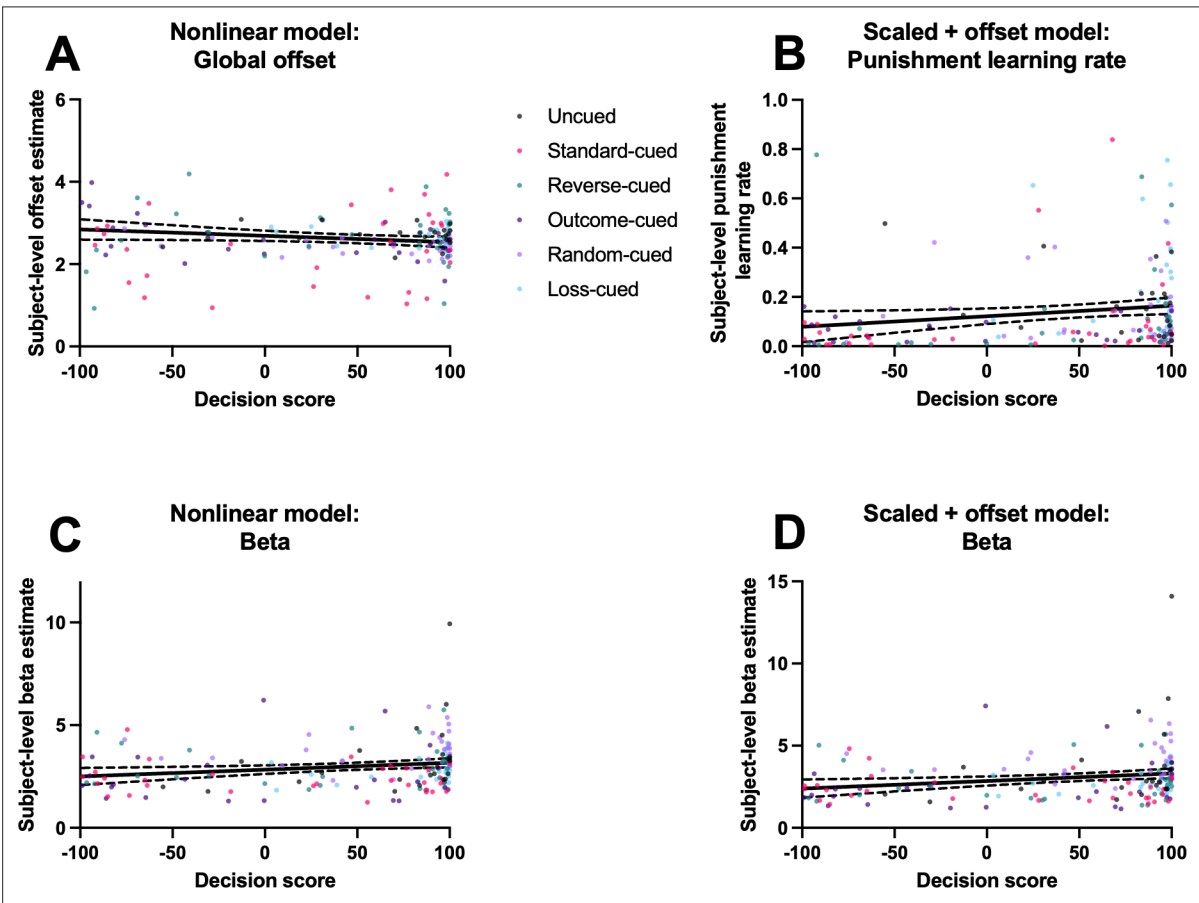

**Figure 8.** Relationships between subject-level parameter estimates and final decision score across rats. Simple linear regression models were fit for each parameter to assess whether parameter values covaried with decision score at the end of training. Dotted lines indicate 95% confidence intervals around the regression lines. (**A**) Subject-level global offset (*b*) estimates from nonlinear model versus final decision score. (**B**) Subject-level punishment learning rate (*η–*) estimates from scaled + offset model versus final decision score. (**C**) Subject-level beta (*β*) estimates from nonlinear model versus final decision score. (**D**) As in (**C**), with beta estimates from the scaled + offset model. N = 165 rats.

These analyses showed that across both task variants, there was evidence of reduced punishment sensitivity (i.e., lower $m$ parameter or punishment learning rate) in risky versus optimal rats. We conducted similar analyses here to examine whether parameter estimates covary with decision score at end of training. To accomplish this, we fit simple linear regression models for each parameter and assessed whether the slopes were significantly different from zero.

While the overall variance explained by these models was modest, we found consistent relationships between parameters controlling sensitivity to punishment and final decision score. For the nonlinear model, we found that the global offset parameter was significantly associated with rats' risk preference in later sessions; negative learning rate reached marginal significance (offset ($b$ parameter): $R^2=0.03$, $F(1, 163)=3.93$, p=0.04; punishment learning rate: $R^2=0.02$, $F(1, 163)=3.34$, p=0.07; *Figure 8A*). For the scaled + offset model, the punishment learning rate was significantly associated with rats' final risk preference ($R^2=0.03$, $F(1, 163)=4.66$, p=0.03; *Figure 8B*). These relationships, while modest in strength, indicate that risky choice was associated with a lower punishment learning rate and a higher global offset of time-out penalty costs, providing further evidence that diminished impact of time-out penalties early in learning can lead to the development of a risky choice profile. Additionally, higher risk preference was associated with a lower beta estimate (nonlinear model: $R^2=0.04$, $F(1, 163)=6.40$, p=0.01; scaled + offset model: $R^2=0.04$, $F(1, 163)=6.96$, p=0.009; *Figure 8C and D*), indicating that risky rats' choices did not follow the latent $Q$-values as closely compared to the optimal rats.

## Discussion

Here, we showed that audiovisual cues drive risky choice on the rGT only if they are reliably, but not exclusively, win-paired. This was demonstrated by higher levels of risky choice in rats trained on the standard-cued, reverse-cued, and outcome-cued variants of the rGT. Computational analysis of the acquisition phase using reinforcement learning models revealed that differences in decision making were largely captured by parameters that control learning from punishments. These parameters predicted decision score at the end of training, indicating that risk-preferring rats discounted losses to a greater degree than optimal rats. There was also evidence that tasks featuring win-paired cues and risky choice in general were associated with lower beta estimates; this may be due to inconsistency in choice patterns early in training, perhaps resulting from reduced sensitivity to outcomes.

These results largely confirm and build upon the previous report investigating learning dynamics of the cued versus uncued rGT (*Langdon et al., 2019*). In order to maximize comparison of multiple cue-outcome schedules, we were unable to also include animals of both sexes, which we acknowledge is a significant limitation that must be addressed in future work. We previously observed that the addition of reward-paired cues to the task resulted in insensitivity to punishments, particularly on the risky options, and that time-out penalty weights could predict decision score at the end of training. The current studies extend these results to suggest that this relationship can be bidirectionally modulated, as loss-paired cues reduce risky choice and increase learning from losing outcomes. Furthermore, pairing cues with wins seems to dominate the decision-making process. Cuing the losses when the wins are also cued has no risk-reducing effect, and in fact, may even further potentiate risky choice.

Results from the reinforcer devaluation test provide additional support for differences in decision-making processes when win-paired cues are present, in that risk-preferring rats trained on these tasks were not sensitive to changes in reinforcer value. This indicates that such cues can render choice patterns inflexible. However, no differences were found between tasks for optimal rats, and they were overall less sensitive to changes in reinforcer value. Optimal rats may therefore be indifferent to fluctuations in reward value such as occasional reward omission on the low-risk options, or in this case, reinforcer devaluation. Additionally, if we assume that frequency of winning becomes more desirable than reward size, optimal rats can only move to P1 to maximize win frequency, whereas risk-preferring rats have more options to which they may shift. Nevertheless, the fact that risk-preferring rats trained on these tasks did not shift suggests that win-paired cues, particularly when they track reward size, inhibit flexible responding in the face of devaluation of those rewards.

While differences in reinforcer devaluation tests in risky versus optimal rats have not been previously observed on the rGT, the cohort sizes of the present study far exceed previous reports, which may have been underpowered to detect such differences. The results from risk-preferring rats corroborate previous studies demonstrating that rats trained on the uncued task are sensitive to this manipulation, whereas rats trained on the cued task are not (*Zeeb and Winstanley, 2013*; *Hathaway et al.,*

*2021*). While risk-preferring rats exhibit some degree of distortion in reward valuation, as they do not follow the most rewarding strategy (i.e., selecting optimal options), we believe this to be at least partially separable from choice inflexibility, as risk-preferring rats on tasks that don't feature win-paired cues remain sensitive to devaluation. Choice inflexibility in these rats could be due to either enhanced habit formation or hypoactive, or otherwise maladaptive, goal-directed control. Altering serotonergic signaling in the lateral orbitofrontal cortex (OFC) can restore sensitivity to reinforcer devaluation in rats trained on the cued rGT, indicating that prefrontal cortices, and presumably impaired goal-directed control, play a role in inflexibility induced by reward-paired cues (*Hathaway et al., 2021*). Indeed, given the role of the lateral OFC in updating stored action-outcome contingencies, cue-guided learning, and in the acquisition of the uncued rGT (*Izquierdo, 2017*; *Amodeo et al., 2017*; *Zeeb and Winstanley, 2011*), this region could be mediating both the differential processing of rewards and punishments across different variants of the rGT and cue-induced inflexibility. Alternatively, the task bracketing hypothesis from *Vandaele et al., 2017* posits that the inclusion of salient cues in decision-making tasks (lever insertion, in their case) encourages the formation of rigid stimulus-response patterns, rather than generally reducing cognitive flexibility. Future studies could test other facets of cognitive flexibility both during and outside of the rGT (e.g., extinction training on the rGT, or probabilistic reversal learning after rGT training) to further explore these lines of thought.

In addition to inducing inflexibility, win-paired cues on the outcome-cued and standard-cued tasks impacted choice patterns in a relatively comparable manner. These results run counter to the 'state' hypothesis described in the introduction, in which we suggested that increasing the similarity of winning and losing trials may permit better integration of the time-out penalties into the learned values of each option. We instead observed that outcome-cued rats were equally as risky as those trained on the standard-cued paradigm. Cuing losses in this paradigm did not increase learning about those trial types; instead, it may have disguised them as wins. Losses disguised as wins (LDWs) are a feature of modern multi-line slot machines in which win-related cues are played when a small payout that is less than the bid is won, leading the player to believe they've won money when in fact they have experienced a net loss. These LDWs can therefore be miscategorized as wins (*Dixon et al., 2010*; *Jensen et al., 2013*). *Marshall and Kirkpatrick, 2017* applied a reinforcement learning model to behavioral data from their task investigating the LDW effect in rodents and showed that playing win-related sensory feedback during losses elevated stay biases on the high-risk option by increasing its value. A similar mechanism may be at play in the outcome-cued task. Conversely, cuing only losses may oppose the LDW effect, as others have shown that playing loss-associated cues during LDWs can permit subjects to correctly categorize them as losses (*Dixon et al., 2015*). It is interesting to note that rats on the outcome-cued task, together with the standard-cued task, had the lowest beta parameter estimate. This may suggest the model did not capture the development of their choice patterns to the same degree as the other tasks. It could be that adding a stay-bias parameter similar to the model by *Marshall and Kirkpatrick, 2017* would better encapsulate the learning dynamics for rats on this task. In general, greater risky choice was predicted by lower beta parameter values across multiple models and task variants. Whether this indicates that risky choice results in part from weaker adherence to internal representations of option values during decision making, or instead suggests we are failing to account for an important computational process in our models remains to be thoroughly investigated.

An alternate hypothesis for the impact of win-paired cues on decision making comes from research investigating the role of dopamine in the perception of time. The timing of dopamine signals can influence whether subjective time speeds up or slows down (*Soares et al., 2016*; *Jakob et al., 2021*). Hence, it could be that cued rewards alter the subjective experience of the time-out penalty duration via a dopaminergic mechanism. Indeed, the standard-cued task is more sensitive to dopamine manipulations than the uncued task (*Barrus and Winstanley, 2016*). Dopamine signals provoked by win-paired cues may reduce the experienced duration of the time-out penalties such that their impact on the latent value of each option is diminished. Measurements of dopamine signals on-task using fiber photometry could be incorporated into a model to test this hypothesis (see *Jakob et al., 2021*).

While pairing cues with wins is sufficient to drive risky choice, cue complexity and magnitude appear to also play a role, as rats on the reverse-cued task were significantly less risky than the outcome-cued rats and marginally different from the standard-cued rats. Additionally, parameter estimates for rats trained on the reverse-cued task did not completely align with the other tasks featuring win-paired cues (e.g., lower learning rate for rewarded trials). These rats also exhibited a lower rate of premature

responding compared to all other tasks except for the loss-cued task. This may indicate that matching cue size and complexity to reward size can potentiate motor impulsivity. Indeed, when the salience of reward-predictive cues matches the size of the reward, activity in the nucleus accumbens is amplified in humans (*Knutson et al., 2001*), which may be diminished when cue size inversely scales with reward. As activity within the nucleus accumbens is critically involved in motor impulsivity on similar behavioral tasks (*Economidou et al., 2012*; *Pattij et al., 2007*), reduction of this signal could explain the low rate of premature responding in these rats.

Consistently pairing cues with wins proved to be a necessary component to induce risky choice, as playing cues randomly on 50% of trials regardless of outcome did not significantly shift risk preference compared to the uncued variant. We originally thought that the increased sensory stimulation from the random cues could increase arousal and therefore risk preference. However, rats instead learned to disregard these cues and were perhaps less engaged in the task, as indicated by the longer latencies to collect reward and reduced levels of premature responding. That being said, this finding does not disprove the hypothesis that increased arousal leads to riskier choice patterns; it may be that the cue-reward relationship increases arousal in a way that random cues cannot. Recent results implicating norepinephrine in cue-induced risky choice would suggest that arousal may contribute to the impact of cues on decision making (*Chernoff et al., 2021*). It would be interesting to pretrain rats on the association between the cues and reward prior to rGT training on the random-cued task; in that case, increased risk preference may be observed. This may represent an intriguing model of the effect of ambient lights and sounds of a casino on gambling behavior.

Pairing cues with losses would also ostensibly increase arousal; however, their behavioral impact was quite distinct from that of win-paired cues. Indeed, rats trained on the loss-cued task were the least risk-preferring out of all the groups, including the uncued task. This would suggest that, while the uncued task is usually regarded as a control for the cued task(s), it may also be a deviation from how optimal rats can be. Indeed, *Langdon et al., 2019* found that all rats across both the uncued and standard-cued task were globally less sensitive to the time-out penalties, and that differences in risk preference arose from the degree of this reduced sensitivity. Conversely, rats on the loss-cued task appear to be more sensitive to losses. Thus, they could be more proactively risk-avoiding, and rats on the uncued task may be more willing to sample from the risky options despite having an overall optimal decision-making profile.

In sum, the results from these studies indicate that outcome-associated cues play a significant role in decision-making processes, and their effect is highly dependent on the outcome type with which they are associated. Differences in choice patterns are largely a result of changes to the relative impact of losses on decision making, as revealed by the effect of different cue paradigms on group-level parameter estimates capturing learning from losses in the tested reinforcement learning models. These analyses demonstrate the power of combining modeling approaches with careful behavioral manipulations to inform our understanding of action selection in complex decision-making scenarios. Furthermore, the findings provide critical insight into the influence of the rich sensory environment in casinos and other forms of gambling, particularly the addictive allure of electronic gaming machines.

## Methods

### Subjects

Subjects were four cohorts of 32–64 male Long Evans rats (Charles River Laboratories, St Constant, QC, Canada) weighing 275–300 g upon arrival to the facility. Then, 1–2 weeks following arrival, rats were food-restricted to 14 g of rat chow per day and were maintained at least 85% body weight of an age- and sex-matched control. Water was available ad libitum. All subjects were pair-housed or trio-housed in a climate-controlled colony room under a 12 h reverse light-dark cycle (21°C; lights off at 8 am). Huts and paper towels were provided as environmental enrichment. Behavioral testing took place 5 days per week. Housing and testing conditions were in accordance with the Canadian Council of Animal Care, and experimental protocols were approved by the UBC Animal Care Committee.

### Behavioral apparatus

Testing took place in 32 standard five-hole operant chambers, each of which was enclosed in a ventilated, sound-attenuating chamber (Med Associates Inc, Vermont). Chambers were fitted with an array

composed of five equidistantly spaced response holes. A stimulus light was located at the back of each hole, and nose-poke responses into these apertures were detected by vertical infrared beams. On the opposite wall, sucrose pellets (45 mg; Bioserv, New Jersey) were delivered to the magazine via an external pellet dispenser. The food magazine was also fitted with a tray light and infrared sensors to detect sucrose pellet collection. A house light could illuminate the chamber. The operant chambers were operated by software written in Med-PC by CAW, running on an IBM-compatible computer.

## Behavioral testing

Rats were first habituated to the operant chambers in two daily 30 min sessions, during which sucrose pellets were present in the nose-poke apertures and food magazine. Rats were then trained on a variant of the five-choice serial reaction time task and the forced-choice variant of the rGT, as described in previous reports (*Zeeb et al., 2009*; *Barrus and Winstanley, 2016*).

A task schematic of the rGT is provided in *Figure 1*. During the 30 min session, trials were initiated by making a nose-poke response within the illuminated food magazine. This response extinguished the light, which was followed by a 5 s inter-trial interval (ITI) in which rats were required to inhibit their responses to proceed with the trial. Any response in the five-hole array during the ITI was recorded as a premature response and punished by a 5 s time-out period, during which the house light was illuminated and no response could be made. Following the ITI, apertures 1, 2, 4, and 5 in the five-hole array were illuminated for 10 s. A lack of response after 10 s was recorded as an omission, at which point the food magazine was re-illuminated and rats could initiate a new trial. A nose-poke response within one of the illuminated apertures was either rewarded or punished according to that aperture's reinforcement schedule. Probability of reward varied among options (0.9–0.4, P1–P4), as did reward size (1–4 sucrose pellets). Punishments were signaled by a light flashing at 0.5 Hz within the chosen aperture, signaling a time-out penalty which lasted for 5–40 s depending on the aperture selected. The task was designed such that the optimal strategy to earn the highest number of sucrose pellets during the 30 min session would be to exclusively select the P2 option, due to the relatively high probability of reward (0.8) and short, infrequent time-out penalties (10 s, 0.2 probability). While options P3 and P4 provide higher per-trial gains of three or four sucrose pellets, the longer and more frequent time-out penalties associated with these options greatly reduce the occurrence of rewarded trials. Consistently selecting these options results in fewer sucrose pellets earned across the session and is therefore considered disadvantageous. The position of each option was counterbalanced across rats to mitigate potential side bias. Half the animals in each project were trained on version A (left to right arrangement: P1, P4, P2, P3) and the other half on version B (left to right arrangement: P4, P1, P3, P2).

### Task variants

Six variants of the task were used in this experiment (n=28–32 rats per task variant). On the uncued task, winning trials were signaled by the illumination of the food magazine alone. On the standard-cued task, reward delivery occurred concurrently with 2 s compound tone/light cues. Cue complexity and variability scaled with reward size, such that the P1 cue consisted of a single tone and illuminated aperture, and the P4 cue consisted of multiple tones and flashing aperture lights presented in four different patterns across rewarded trials. The reverse-cued task featured an inversion of the cue-reward size relationship, such that the longest and most complex cue occurred on P1 winning trials, and P4 winning trials were accompanied by a single tone and illuminated aperture. On the outcome-cued task, all trial outcomes were accompanied by an audiovisual cue (i.e., during reward delivery and at the onset of the time-out penalty). The random-cued task consisted of cues that occurred on 50% of trials, regardless of outcome. Lastly, on the loss-cued task, cues occurred only on losing trials, at the onset of the time-out penalty. Cue complexity and magnitude scaled with reward size/time-out penalty length for the outcome-, random-, and loss-cued variants of the task (i.e., same pattern as the standard-cued task).

## Reinforcer devaluation

126 rats (n=12–28 per task version) underwent a reinforcer devaluation procedure. This procedure took place across 2 days. On the first day, half of the rats were given ad libitum access to the sucrose pellets used as a reward on the rGT for 1 h prior to task initiation. The remaining rats completed the rGT without prior access to sucrose pellets. Following a baseline session day for which no sucrose

pellets were administered prior to the task to any rats, the groups were then reversed and the other half were given 1 h access to sucrose pellets.

## Behavioral measures and data analysis

All statistical analyses were completed using SPSS Statistics 27.0 software (SPSS/IBM, Chicago, IL, USA) and the ArviZ Python package (*Kumar et al., 2019*). As per previous reports, the following rGT variables were analyzed: percentage choice of each option ([number of times option chosen/ total number of choices]×100), decision score (calculated as percent choice of [(P1+P2) − (P3+P4)]), percentage of premature responses ([number of premature responses/total number of trials initiated]×100), sum of omitted responses, sum of trials completed, and average latencies to choose an option and collect reward. Variables that were expressed as a percentage were subjected to an arcsine transformation to limit the effect of an artificially imposed ceiling (i.e., 100%). Animals with a mean positive baseline decision score were designated as 'optimal', whereas rats with negative decision scores were classified as 'risk-preferring'.

For baseline analyses, mean values for each variable were calculated by averaging across four consecutive sessions that were deemed statistically stable (i.e., session and/or session × choice inter- action were not significant in a repeated-measures ANOVA; following approximately 35–40 training sessions). Task (six levels: uncued, standard-cued, reverse-cued, outcome-cued, random-cued, loss- cued) and risk status (two levels: optimal, risk-preferring) were included as between-subjects factors for all baseline analyses. Choice data were analyzed with a two-way repeated measures ANOVA with choice (four levels: P1, P2, P3, and P4) as within-subject factors. For the analysis of the reinforcer devaluation data, devaluation (two levels: baseline, devaluation) and choice (four levels: P1–P4) were the within-subject factors and task version and risk status were the between-subjects factors.

For all analyses, if sphericity was violated as determined by Mauchley's test, a Huynh–Feldt correc- tion was applied, and corrected p values' degrees of freedom were rounded to the nearest integer. Results were deemed to be significant if p values were less than or equal to an α of 0.05. Any main effects or interactions of significance were further analyzed via post hoc one-way ANOVA or Tukey's tests. Any p values > 0.05 but <0 .09 were reported as a statistical trend.

## Hierarchical modeling of learning from wins and losses

A full description of the modeling approach can be found in *Langdon et al., 2019*. Valid choice trials from the first five sessions were concatenated into one long session and trial-by-trial preferences were modeled using variations on the Q-learning algorithm from reinforcement learning (RL; *Sutton and Barto, 1998*). Each model was fit separately to each task variant group, thus allowing for the possi- bility that different RL models might perform better at predicting choice for each of the groups. Data from 11 rats were excluded due to missing sessions or technical issues. This left a total of 24 rats in the uncued task group, 32 rats in the standard-cued task group, 25 rats in the outcome-cued task group, and 28 rats in the reverse-, random-, and loss-cued task groups.

Each of these models assumes that choice on every trial probabilistically follows latent $Q$-values for each option, and these are updated iteratively according to the experienced outcomes. For our models, the probability of choosing option $P_x$ on each trial follows the learned $Q$-values for $x = [1,2,3,4]$ according to the softmax decision rule:

$$p\left(P_x\right) = \frac{e^{\beta Q_x}}{\sum_{y=1}^{4} e^{\beta Q_y}},$$

where $p(P_x)$ is the probability of choosing option $P_x$, $Q_x$ is the learned latent value of option $x$, and $\beta$ is the inverse temperature parameter that controls how strongly choice follows the latent $Q$-values rather than a random (uniform) distribution over the four options. In each learning model, we assume learning of latent $Q$-values from positive outcomes follows a simple delta-rule update:

$$Q_x^{new} = Q_x^{old} + \eta^+ \left(R_{tr} - Q_x^{old}\right),$$

where $\eta^+$ is a learning rate parameter that governs the step-size of the update, $R_{tr} > 0$ is the number of pellets delivered on a given winning trial, and $Q_x$ is the latent value for the chosen option $x$ on a given trial.

Q-values for learning from punishments were updated differently depending on the model. In each case, we sought to model the negative impact of time-out penalties on choice by transforming the duration of the penalty into an equivalent 'cost' in sucrose pellets. The most basic model uses the same delta-rule update for losing trials, with a separate punishment learning rate ($\eta^-$):

$$Q_x^{new} = Q_x^{old} + \eta^- \left( 0 - Q_x^{old} \right)$$

In this model, time-out penalties do not explicitly influence the trial-wise decrease in Q-values on loss trials. Thus, three different models were designed to determine how time-out penalty duration impacted Q-values. Each model tests a different hypothesis on the transform of the punishments, with a separate punishment learning rate $\eta-$.

| Model name | Parameters | Punishment update |
|---|---|---|
| Scaled cost | 4 | $Q_x^{new} = Q_x^{old} + \eta^- \left( -mT_{tr} - Q_x^{old} \right)$ |
| Scaled + offset cost | 5 | $Q_x^{new} = Q_x^{old} + \eta^- \left( b - mT_{tr} - Q_x^{old} \right)$ |
| Nonlinear cost | 5 | $Q_x^{new} = Q_x^{old} + \eta^- \left( b - T_{tr}^r - Q_x^{old} \right)$ |

In the scaled punishment model, we assume that the equivalent punishment for a time-out penalty on each losing trial scales linearly with the duration of the punishment. $T_{tr} > 0$ is the time-out penalty duration in seconds on a given losing trial and $m$ is a scaling parameter that maps time-out duration into an equivalent cost in pellets (i.e., has units pellets/s). The scaled + offset model is the same as the scaled punishment model but features an additional offset parameter $b$, which removes the constraint that the linear transform between time-out penalty duration and equivalent cost is zero for zero duration.

An independent cost model, as originally described in **Langdon et al., 2019**, was initially used to model a nonlinear relationship between penalty duration and equivalent cost. In this model, equivalent costs for each option are controlled independently by $\omega_x$ for each option $P_x$. The qualitative effects were still present in these model fits. However, due to the higher degree of model complexity and smaller datasets featured here (24–32 rats) versus datasets in **Langdon et al., 2019**; >100 rats, independent parameters were not well isolated. Thus, we developed a more constrained nonlinear model, which uses a power function to allow a nonlinear mapping between experienced duration and the equivalent cost in pellets on each trial. The curvature of this relationship is determined by a single parameter $r$. This function has been previously used to describe expected utility during risky choice (**Holt and Laury, 2002**; **Lopez-Guzman et al., 2018**).

For every model, Q-values were initialized at zero for the first session, and we assumed Q-values at the start of a subsequent session (on the next day, for example) were the same as at the end of the previous session (i.e., we modeled no intersession effects on learning). Each model was fit to the entire set of choices for each group of rats using Hamiltonian Monte Carlo sampling with Stan to perform full Bayesian inference and return the posterior distribution of the model parameters conditional on the data and specification of the model (**Carpenter et al., 2017**). In each case, we partially pooled choice data across individual rats in a hierarchical model to simultaneously determine the distribution of individual- and group-level model parameters. We implemented a noncentered parameterization for group-level β, $\eta+$, and $\eta-$ in each model, as this has been shown to improve performance and reduce autocorrelation between these group-level parameters in hierarchical RL models (**Ahn et al., 2017**).

Each model was fit using four chains with 1000 steps each (after an initial 1000 burn-in), yielding a total of 4000 posterior samples. To assess the convergence of the chains, we computed the $R$ statistic (**Gelman et al., 2013**), which measures the degree of variation between chains relative to the variation within chains. If the $R$ statistic exceeded 1.01, the number of warmup iterations was increased to a maximum of 5000. Using this approach, across all three models, no group-level or subject-level parameter had $R > 1.01$, and the mode was 1.00, indicating that for each model all chains had converged successfully.

To measure the difference between group-level parameters, we used highest density intervals (HDI; *Kruschke, 2014*). The HDI is the interval that contains the required mass such that all points within the interval have a higher probability density than points outside the interval. Differences were considered credible when the 95% HDI for the sample difference between two mean estimates did not include zero. To compare the overall performance of each model, we computed the Watanabe–Akaike information criterion (WAIC; *Watanabe, 2010*), which, like AIC or BIC, provides a metric to compare different models fit to the same dataset. The WAIC is computed from the pointwise log-likelihood of the full posterior distribution (thereby assessing model fit) with a second term penalizing for model complexity.

## Acknowledgements

This work was supported by a Discovery Grant awarded to CAW from the Natural Sciences and Engineering Council of Canada (NSERC; RGPIN-2017-05006) and a project grant (PJT-162312) awarded to CAW from the Canadian Institutes of Health Research (CIHR). BAH was supported by a CIHR Doctoral Award, and DRK was supported by an Undergraduate Summer Research Award from NSERC. AJL and BAH were supported by the Intramural Research Program at the National Institute of Mental Health (ZIA-MH002983). The experimental work took place at a UBC campus situated on the traditional, ancestral, and unceded land of the xʷməθkʷəy̓əm (Musqueam), səĺílwətaʔɬ/Selilwitulh (Tsleil-Waututh) and Sḵwx̱wú7mesh (Squamish) Peoples. We acknowledge and are grateful for their stewardship of this land for thousands of years.

## Additional information

### Funding

| Funder | Grant reference number | Author |
|---|---|---|
| Natural Sciences and Engineering Research Council of Canada | RGPIN-2017-05006 | Catharine Winstanley |
| Canadian Institutes of Health Research | PJT-162312 | Catharine Winstanley |
| National Institute of Mental Health | ZIA-MH002983 | Angela Langdon |

The funders had no role in study design, data collection and interpretation, or the decision to submit the work for publication.

### Author contributions

Brett A Hathaway, Conceptualization, Data curation, Software, Formal analysis, Investigation, Visualization, Methodology, Writing – original draft, Writing – review and editing; Dexter R Kim, Data curation, Software, Formal analysis, Visualization; Salwa BA Malhas, Data curation, Investigation; Kelly M Hrelja, Investigation, Project administration; Lauren Kerker, Data curation, Software, Formal analysis; Tristan J Hynes, Project administration; Celyn Harris, Investigation; Angela Langdon, Software, Formal analysis, Funding acquisition, Methodology, Writing – review and editing; Catharine Winstanley, Conceptualization, Resources, Software, Supervision, Funding acquisition, Methodology, Writing – review and editing

### Author ORCIDs

Brett A Hathaway ⬤ https://orcid.org/0000-0002-1905-2199
Angela Langdon ⬤ https://orcid.org/0000-0003-4742-6976
Catharine Winstanley ⬤ https://orcid.org/0000-0001-7032-4471

### Ethics

All procedures were conducted in accordance with the guidelines of the Canadian Council on Animal Care. Experimental protocols (A21-0012) were reviewed and approved by the University of British Columbia Animal Care Committee. Animals were monitored regularly, and all efforts were made to minimize stress and discomfort.

Reviewer #2 (Public review): https://doi.org/10.7554/eLife.105951.3.sa1
Reviewer #3 (Public review): https://doi.org/10.7554/eLife.105951.3.sa2
Author response https://doi.org/10.7554/eLife.105951.3.sa3

# Additional files

## Supplementary files
Supplementary file 1. Additional statistical comparisons across task variants.

MDAR checklist

## Data availability
The behavioral data and custom code used for model fitting and analysis are openly available via the Open Science Framework (OSF) at https://doi.org/10.17605/OSF.IO/PDG85.

The following dataset was generated:

| Author(s) | Year | Dataset title | Dataset URL | Database and Identifier |
| --- | --- | --- | --- | --- |
| Hathaway BA, Kim DR, Malhas SBA, Hrelja KM, Kerker L, Hynes TJ, Harris C, Langdon AJ, Winstanley CA | 2022 | Data from: Audiovisual cues must be predictable and win-paired to drive risky choice | https://doi.org/10.17605/OSF.IO/PDG85 | Open Science Framework, 10.17605/OSF.IO/PDG85 |

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
