## [Editor Report · eLife Assessment]

This **important** study provides a nuanced analysis of the impact of cues on cost/benefit decision-making deficits in male rats that could have translational relevance to many addictive disorders. The main findings are that cues paired with rewarded outcomes increase the proportion of risky outcomes, whereas risky choice is reduced when cues are paired with reward loss. The experimental data are **compelling**, whereas the computational analysis based on the optimization of different Q-learning models is **solid**. The findings will be of interest to behavioral neuroscientists and clinicians with an interest in risk, decision making, and gambling disorders.

---

## [Referee Report · Reviewer #2 (Public review)]

Summary:

The manuscript by Hathaway et al. describes a set of elegant behavioral experiments designed to understand which aspects of cue-reward contingencies drive risky choice behavior. The authors developed several clever variants of the well established rodent gambling task (also developed by this group) to understand how audiovisual cues alter learning, choice behavior, and risk. Computational and sophisticated statistical approaches were used to provide evidence that: (1) audiovisual cues drive risky choice if they are paired with rewards and decrease risk if only paired with loss, (2) pairing cues with rewards reduces learning from punishment, and (3) differences in risk taking seem to be present early on in training.

Strengths:

The paper is well written, the experiments well designed, and the results are highly interesting particularly for understanding how cues can motivate and invigorate normal and abnormal behavior.

Comments on revisions:

The authors have done an exceptional job at addressing my initial concerns and questions regarding the evidence to support their claims. I have no additional suggestions or concerns.

---

## [Referee Report · Reviewer #3 (Public review)]

Summary:

In this work, Hathaway and colleagues aim to understand how audiovisual cues at time of outcome promote selection of risky choices. A real life illustration of this effect is used in electronic gambling machines which signal a win with flashing lights and jingles, encouraging the player to keep betting. More specifically, the author asks whether the cue has to be paired exclusively to wins, or whether it can be paired to both outcomes, or exclusively loss outcomes, or occur randomly. To tackle this question, they employ a version of the Iowa Gambling Task adapted to rats, and test the effect of different rules of cue-outcome associations on the probability of selecting the riskier options; they then test the effect of prior reward devaluation on the task; finally, the optimise computational models on the early phases of the experiment to investigate potential mechanisms underlying the behavioural differences.

Strengths:

The experimental approach is very thorough, in particular the choice of the different task variants cover a wide range of different potential hypotheses. Using this approach, they find that, although rats prefer the optimal choices, there is a shift towards selecting riskier options in the variants of the task where the cue is paired to win outcomes. They analyse this population average shift by showing that there is a concurrent increase in the number of risk-taking individuals in these tasks. They also make the novel discovery that pairing cues with loss outcomes instead reduces the tendency for risky decisions.

The computational strategy is appropriate and in keeping with the accepted state of the art: defining a set of candidate models, optimising them, comparing them, simulating the best ones to ensure they replicate the main experimental results, then analysing parameter estimates in the different tasks to speculate abut potential mechanisms.

Weaknesses:

While the overall computational approach is excellent, there is a missed opportunity in the computational modelling section due to the choice of models which is dependent on a preceding study by Langdon et al. (2019). Loss trials come at a double cost: firstly the lost opportunity of not having selected a winning option which is reflected straightforwardly in Q-learning by the fact that r=0, secondly a waiting period which will affect the overall reward rate. The authors combine these costs by converting the time penalty into "reward currency" using three different functions which make up the three different tested models. This means the question when comparing models is not something along the lines of "are individuals in the paired win-cue tasks more sensitive to risk? or less sensitive to time? etc." but rather "what is the best way of converting time into Q-value currency to fit the data?". Instead, the authors could have contrasted other models which explicitly track time as a separate variable (see for example "Impulsivity and risk-seeking as Bayesian inference under dopaminergic control" (Mikhael & Gershman 2021)) or give actions an extra risk bonus (as in "Nicotinic receptors in the VTA promote uncertainty seeking" (Naude et al 2016)) to better disentangle the mechanisms at play.

---

## [Author Response]

The following is the authors’ response to the original reviews

Public Reviews:

**Reviewer 1 (Public review):**
When do behavioral differences emerge between the task variants? Based on the results and discussion, the cues increase the salience of either the wins or the losses, biasing behavior in favor of either risky or optimal choice. If this is the case, one might expect the cues to expedite learning, particularly in the standard and loss condition. Providing an analysis of the acquisition of the tasks may provide insight into how the cues are "teaching" decision-making and might explain how biases are formed and cemented.

While considerable differences in decision making emerge in early sessions of training, we do not observe any evidence that cuing outcomes expedites the development of stable choice patterns. Indeed, since the outcomes are cued across all four options, there is no categorical difference in salience between optimal and risky choices. Thus, our interpretation is that cuing wins and/or losses alters the integration of this feedback into choice preference, rather than the rate of the development of choice preference. To quantitatively address this point, we have included the following analysis:

“To quantitatively examine choice variability during training, we binned sessions 1-5 and 6-10 and analyzed variability in choice patterns across task variants. Analysis of the first five sessions of training revealed a significant shift in decision score across sessions (F(3, 502) = 31.23, p <.0001), which differed between task variants (session x task: F(16, 502) = 2.13, p = .007). Conversely, while significant differences in overall score were observed between task variants in sessions 6-10 (task: F(5, 156) = 6.81, p <.0001), there was no significant variability across sessions (session: F(3, 481) = 2.06, p = .10, task x session: F(15, 481) = 0.78, p = .71). This indicates that the variability in choice preference (and presumably, learning about outcomes) is maximized in the first five sessions, and there are no obvious differences in the rate of development of stable choice patterns between task variants.”

Does the learning period used for the modeling impact the interpretation of the behavioral results? The authors indicate that computational modeling was done on the first five sessions and used these data to predict preferences at baseline. Based on these results, punishment learning predicts choice preference. However, these animals are not naïve to the contingencies because of the forced choice training prior to the task, which may impact behavior in these early sessions. Though punishment learning may initially predict risk preference, other parameters later in training may also predict behavior at baseline.

The first five sessions were chosen based on a previously developed method used in Langdon et al. (2019). When choosing the number of sessions to include, there is a balance between including more data points to improve estimation of parameters while also targeting the timeframe of maximal learning. As training continues, the impact of outcomes on subsequent choice should decrease, and the learning rate would trend towards zero. This can be observed in the reduction in inter-session choice variability as training progresses, as demonstrated in the analyses above. Once learning has ceased, presumably other cognitive processes may dictate choice (for example, habitual stimulus-response associations), which would not be appropriately captured by reinforcement learning models. It would be a separate research question to determine the point at which parameters no longer become predictive, requiring a larger dataset to thoroughly assess. We acknowledge that we did not provide sufficient justification for the learning period used for the modeling. In conjunction with the analysis of early sessions outlined above, we have added the following to the text:

“We investigated differences in the acquisition of each task variant by fitting several reinforcement learning (RL) models to early sessions. Our modeling approach closely follows methods outlined in Langdon et al. (2019), in which a much larger dataset (>100 rats per task) was used to develop the RL models applied here. Due to the comparatively small *n* per group in the current study, we limited our model selection to those previously validated in Langdon et al. (2019), with minor extensions. As in previous work, models were fit to valid choices from the first five sessions. As training continues, the impact of outcomes on subsequent choice should decline, and parameter values may evolve over time (e.g., decreasing learning rate). To target the period of learning during which outcomes have maximal influence over choice, and parameters likely have fixed values, we limited our analyses to the first five sessions.”

The authors also present simulated data from the models for sessions 18-20, but according to the statistical analysis section, sessions 35-40 were used for analysis (and presumably presented in Figure 1). If the simulation is carried out in sessions 35-40, do the models fit the data?

Based on our experience, choice patterns are well instantiated by session 20, and training only continues to 30+ sessions to achieve stability in other task variables (e.g., latencies, premature responding, etc.). That being said, the discrepancy between session numbers is confusing, so we’ve extended the simulations to match the same session numbers that were analyzed in the experimental data.

Finally, though the n's are small, it would be interesting to see how the devaluation impacts computational metrics. These additional analyses may help to explain the nuanced effects of the cues in the task variants.

Unfortunately, as the devaluation experiment is only one session, there are insufficient data to run the same models. Furthermore, changes in choice are subtle and not uniform across rats, making it difficult to reliably model this effect at the individual level. A separate experiment could investigate the specific cognitive processes underlying the devaluation effect.

**Reviewer #1 (Recommendations for the authors):**
The authors do not present individual data points for behavior. Including these data points would improve the interpretability of the results. Adding significant notations to the bar graphs would also help the reader. Although the stats are provided and significant comparisons highlighted, it isn't easy to go between the table and the figure to detect significant outcomes. If done, the statistics tables could be moved to the supplement. Including estimates of effect size for main findings in the main text would also benefit the reader.

We thank the reviewer for their feedback on our approach to the figures and significance reporting – we have updated the relevant figures to include individual data points. Furthermore, we’ve added significance notations for task variants that are significantly different from the uncued or standard cued tasks on the figures. We’ve also moved some statistics tables to the supplement, as suggested.

The authors allude to other metrics of the task (trials, omissions, etc.) but do not present these data anywhere. Including supplementary figures including individual data points and statistical analyses in the supplement is strongly encouraged.

A supplementary figure visualizing these metrics (choice latency, trials completed, and omissions) has been added, with individual data points included. Statistical analyses are reported in the main text – no significant effect in the ANOVAs were observed for any of these metrics, so post hoc analyses were not performed.

Figure 4 is confusing. Presenting the WAIC values for each model rather than compared to the nonlinear model would be easier to understand. It is also unclear if statistical tests were used to assess differences in model fit as no test information is provided.

Figure 4 has been updated to increase clarity and address feedback from another reviewer. Raw WAIC values are not ideal for visualization, as the task variants have differing amounts of data and thus would be difficult to include on the same Y-axis. Instead, we present each model’s difference in WAIC relative to a basic model with no timeout penalty transform, so that all three models are visible, and the direction of model improvement is clearly indicated. Statistical tests of WAIC differences are not standard, as the numerical differences themselves indicate a better fit.

The authors do not provide a data availability statement.

We thank the reviewer for calling our attention to this oversight. A data availability statement has been added.

**Reviewer 2 (Public review):**
Additional support and evidence are needed for the claims made by the authors. Some of the statements are inconsistent with the data and/or analyses or are only weakly supportive of the claims.

We appreciate the reviewer’s overarching concern that some claims in the original manuscript were insufficiently supported by the data or analyses. To address this, we have provided further rationale for the devaluation experiment and clarified our interpretation of those results, expanded the computational modeling analyses, and revised figures and wording to improve clarity. Below, we respond to the reviewer’s specific comments in detail.

**Reviewer #2 (Recommendations for the authors):**
Different variants of an RL model were used to understand how loss outcomes impacted choice behavior across the gambling task variants. Did the authors try different variants for rewarded outcomes? I wonder whether the loss specific RL effects are constrained to that domain or perhaps emerged because choice behavior to losses was better estimated with the different RL variants. For example, rewarded outcomes across the different choices may not scale linearly (e.g., 1, 2, 3, 4) so including a model in which Rtr is scaled by a free parameter might improve the fit for win choices.

We agree that asymmetries in model flexibility could, in principle, contribute to the observed effects. While we are somewhat limited in our ability to develop and validate further models due to the small size of the datasets compared to the high degree of choice variability between rats, we have explored the possibility as far as the data allow by fitting a model that includes a scaling parameter for rewards in addition to punishments:

“While we restricted our model selection to those previously validated on larger datasets, the specificity of the main finding to the punishment learning rate may be due to the greater flexibility afforded to loss scaling, rather than a true asymmetry in learning. To test this hypothesis, we fit a model featuring a scaling parameter for rewards, in addition to scaled costs:\begin{document}$$\displaystyle Q_{x}^{\mathrm{new}}=Q_{x}^{\mathrm{old}}+\eta^{+}\left(mRewxR_{tr}-Q_{x}^{\mathrm{old}}\right)$$\end{document}

where *mRew* is a linear scaling parameter for reward size. A separate scaling parameter was used for timeout penalty duration (i.e., same as scaled cost model). Group-level parameter estimates (Figure S3) reflected similar differences in the punishment learning rate and reward learning rate as the scaled cost model (Figure S4). Furthermore, all 95% HDIs for the *mRew* scaling parameter included 1, indicating that at least at the group level, scaling of reward size across the P1-P4 options closely follows the actual number of earned sucrose pellets. Thus, we find no evidence that our results can be simply attributed to the increased parameterization of losing outcomes.”

Additionally, I would like to see evidence that these alternative models provide a better fit compared to a standard delta-rule updating for unrewarded choices.

Each model is now compared directly to a standard delta-rule update model in the WAIC figure to demonstrate that the current models are a better fit for the data.

Could the authors provide some visualization of how variation in the r, m, or b parameters impact choices and/or patterns of choices?

We have added a figure to the supplementary section to visualize how different values for the *r*, *m*, and *b* parameters could alter the size of updates to Q-values on each trial across the four different options, thereby impacting subsequent choice.

It was challenging to understand the impact of the reported effects and interpretation of the authors at various points in the manuscript. For example, the authors state that "only rats trained on tasks without win-paired cues exhibited shifts in risk preference following reinforcer devaluation". Figure 3 however seems to indicate that rats trained on the reverse-cued task show shifts in risk preference.

We agree the original wording did not fully capture the nuance apparent in the figure. While not significantly different from baseline, rats in the reverse-cued experiment could have indeed updated their choice patterns and we were underpowered to detect the effect. We have updated the results section to include this point, and to more specifically outline that win-paired cues that scale with reward size lead to insensitivity to reinforcer devaluation:

“This indicates that pairing audiovisual cues with reward induces some degree of inflexibility in risk-preferring rats. Importantly, pairing cues with losses alone does not elicit rigidity in choice. Thus, in keeping with the observed effect on overall choice patterns, pairing cues with wins has a unique impact on sensitivity to reinforcer devaluation. Although not statistically significant, visual inspection of the reverse-cued task suggests that some choice flexibility may be present, and the study may be underpowered to detect this effect. Nonetheless, win-paired cues that scale with reward size reduce flexibility in choice patterns following reinforcer devaluation.”

It was not clear to me why the authors did a devaluation test and what was expected. Adding details regarding the motivation for specific analyses and/or experiments would improve understanding of these exciting results.

Further explanation has been added to the results section for the devaluation test to clarify the rationale and expected results:

“We next tested whether pairing salient audiovisual cues with outcomes on the rGT impacts flexibility in decision making when outcome values are updated. Reinforcer devaluation, in which subjects are sated on the sugar pellet reinforcer prior to task performance (presumably devaluing the outcome), is a common test of flexibility of decision making (Adams & Dickinson, 1981). We have previously employed this method to demonstrate that rats trained on the standard-cued task are insensitive to reinforcer devaluation (i.e., choice patterns do not shift despite devaluation of the sugar pellet reward; Hathaway et al., 2021).”

Some rats in the rGT become risk takers and some do not, but whether this is an innate phenomenon or emerges with training is not known. The authors report some correlations between the RL parameters and subsequent risk scores but this may be an artifact because the risk scores and many of the parameters differ between the experimental groups. Restricting these analyses to the rats in the standard procedure (or even conducting it in other rats that have been run in the rGT standard task) would alleviate this concern. The authors should also expand upon this result in the discussion. (if it holds up) and provide graphs of this relationship in the manuscript.

In a previous paper on which these analyses were based (Langdon et al., 2019), analyses of the relationship between RL parameter estimates and final decision score were conducted separately for rats trained on either the uncued or standard cued task, as the reviewer has suggested here. Those analyses showed that parameters controlling the learning from negative outcomes were specifically related to final score in both tasks. While we don’t have the appropriate *n* per group to split the analyses by task variant in the current study, we have highlighted these previous findings in the results section to address this concern:

“In Langdon et al. (2019), analyses were conducted to test whether parameters controlling sensitivity to punishment predicted final decision score at the end of training in the uncued and standard cued task variants. These analyses showed that across both task variants, there was evidence of reduced punishment sensitivity (i.e., lower m parameter or punishment learning rate) in risky versus optimal rats. We conducted similar analyses here to examine whether parameter estimates covary with decision score at end of training. To accomplish this, we fit simple linear regression models for each parameter and assessed whether the slopes were significantly different from zero.”

I don't see a b parameter in the nonlinear cost model, but is presented in Figure 6 and also in the "Parameters predicting risk preference on the rGT". The authors either need to update the formula or clarify what the b parameter quantifies in the nonlinear model.

We thank the reviewer for pointing out this oversight; the equation has been updated to include the b parameter.

The risk score is very confusing as high numbers or % indicate less risk and lower (more negative numbers) indicate greater risk. I've had to reread the text multiple times to remind myself of this, so I anticipate the same will be true for other readers. Perhaps the authors can add a visual guide to their y-axis indicating more positive numbers are less risky choices.

We acknowledge that this measure can be confusing – the calculation of this score is standard for the Iowa Gambling Task conducted in humans, on which the rGT is based, and was therefore adopted here. We’ve changed the name from “risk score” to “decision score”, along with including a visual guide to the y-axis in Figure 2, to address this point.

Negative learning rate is confusing as it almost implies that the learning was a negative value, rather than being a learning rate for negative outcomes. Please revise in the figures and in the text.

We have updated the text and figures where appropriate from “negative learning rate” to “punishment learning rate”. We have also changed the text from “positive learning rate” to “reward learning rate” to match this terminology.

**Reviewer 3 (Public review):**
There is a very problematic statistical stratagem that involves categorising individuals as either risky or optimal based on their choice probabilities. As a measurement or outcome, this is fine, as previously highlighted in the results, but this label is then used as a factor in different ANOVAs to analyse the very same choice probabilities, which then constitutes a circular argument (individuals categorised as risky because they make more risky choices, make more risky choices...).

Risk status was included as a factor to test whether the effects of the cue paradigms differed between risky versus optimal rats (i.e., interaction effects), not as an independent predictor of choice preference. We focus on results showing a significant task x risk status interaction, and conducted follow-up analyses separately within each group, at which point risk status was no longer included as a factor. We do not interpret main effects or choice x status interactions, which would indeed be circular for the reason noted by the reviewer.

A second experiment was done to study the effect of devaluation on risky choices in the different tasks. The results, which are not very clear to understand from Figure 3, would suggest that reward devaluation affects choices in tasks where the win-cue pairing is not present. The authors interpret this result by saying that pairing wins with cues makes the individuals insensitive to reward devaluation. Counter this, if an individual is prone to making risky choices in a given task, this points to an already distorted sense of value as the most rewarding strategy is to make optimal non-risky choices.

We have included significance notations in Figure 3 and included further detail in the text to improve clarity of the findings for the devaluation test. The reviewer raises an interesting point that risk-preferring rats have a distorted sense of value, since they do not follow the optimal strategy. However, we believe that this is at least partially separable from insensitivity to devaluation, since risk-preferring rats trained on tasks that don’t feature win-paired cues still exhibit flexibility in choice. We have added the following point to the discussion to address this:

“While risk-preferring rats exhibit some degree of distortion in reward valuation, as they do not follow the most rewarding strategy (i.e., selecting optimal options), we believe this to be at least partially separable from choice inflexibility, as risk-preferring rats on tasks that don’t feature win-paired cues remain sensitive to devaluation.”

While the overall computational approach is excellent, I believe that the choice of computational models is poor. Loss trials come at a double cost, something the authors might want to elaborate more upon, firstly the lost opportunity of not having selected a winning option which is reflected in Q-learning by the fact that r=0, and secondly a waiting period which will affect the overall reward rate. The authors choose to combine these costs by attempting to convert the time penalty into "reward currency" using three different functions that make up the three different tested models. This is a bit of a wasted opportunity as the question when comparing models is not something like "are individuals in the paired win-cue tasks more sensitive to risk? or less sensitive to time? etc" but "what is the best way of converting time into Q-value currency to fit the data?" Instead, the authors could have contrasted other models that explicitly track time as a separate variable (see for example "Impulsivity and risk-seeking as Bayesian inference under dopaminergic control" (Mikhael & Gershman 2021)) or give actions an extra risk bonus (as in "Nicotinic receptors in the VTA promote uncertainty seeking" (Naude et al 2016)).

We thank the reviewer for their thoughtful suggestions and agree that alternative modeling frameworks that explicitly track time or incorporate uncertainty bonuses would be highly informative for understanding the mechanisms underlying risky choice. However, the models employed here are drawn from previous work that required >100 rats per group for model development and validation. Due to the high degree of variability in decision making within the groups and the relatively small number of rats, this dataset is not well suited for substantial model innovation. Indeed, the most complex model from previous work had to be simplified to achieve model convergence. Testing models that greatly diverge from the previously validated RL models would make it difficult to determine whether poor model fit reflects a misspecified model or insufficient data.

We’d also like to note that the driving question for this study is to investigate the impact of different cue variants on choice patterns – untangling the relationship between timing, uncertainty, and risky choice is an important and interesting question, but beyond the scope of the present work.

To address this limitation, we have expanded our justification of model choice in the results section to emphasize that we are applying previously developed models, with minor extensions:

“We investigated differences in the acquisition of each task variant by fitting several reinforcement learning (RL) models to early sessions. Our modeling approach closely follows methods outlined in Langdon et al. (2019), in which a much larger dataset (>100 rats per task) was used to develop the RL models applied here. Due to the comparatively small n per group in the current study, we limited our model selection to those previously validated in Langdon et al. (2019), with minor extensions.”

Another weakness of the computational section is the fact, that despite simulations having been made, figure 5 only shows the simulated risk scores and not the different choice probabilities which would be a much more interesting metric by which to judge model validity.

We have expanded Figure 5 to show the simulated choice of each option.

In the last section, the authors ask whether the parameter estimates (obtained from optimisation on the early sessions) could be used to predict risk preference. While this is an interesting question to address, the authors give very little explanation as to how they establish any predictive relationship. A figure and more detailed explanation would have been warranted to support their claims.

We have expanded this section to provide clearer detail on the methods used to conduct this analysis and added a figure. To address a point raised by another reviewer, the statistical approach has been revised to more closely align with that used in Langdon et al. (2019), and the results have been updated appropriately:

“We next tested whether any of the subject-level parameter estimates in the nonlinear or scaled + offset model could reliably predict risk preference scores at the end of training. In Langdon et al. (2019), analyses were conducted to test whether parameters controlling sensitivity to punishment predicted final decision score at the end of training in the uncued and standard cued task variants. These analyses showed that across both task variants, there was evidence of reduced punishment sensitivity (i.e., lower *m* parameter or punishment learning rate) in risky versus optimal rats. We conducted similar analyses here to examine whether parameter estimates covary with decision score at end of training. To accomplish this, we fit simple linear regression models for each parameter and assessed whether the slopes were significantly different from zero.”

Why were the simulated risk scores calculated for sessions 18-20 and not 35-39 as in the experimental data, and why were the models optimised only on the first sessions?

These points were addressed in response to reviewer #1:

Based on our experience, choice patterns are well instantiated by session 20, and training only continues to 30+ sessions to achieve stability in other task variables (e.g., latencies, premature responding, etc.). That being said, the discrepancy between session numbers is confusing, so we’ve extended the simulations to match the same session numbers that were analyzed in the experimental data.

The first five sessions were chosen based on a previously developed method used in Langdon et al. (2019). When choosing the number of sessions to include, there is a balance between including more data points to improve estimation of parameters while also targeting the timeframe of maximal learning. As training continues, the impact of outcomes on subsequent choice should decrease, and the learning rate would trend towards zero. This can be observed in the reduction in inter-session choice variability as training progresses, as demonstrated in the analyses above. Once learning has ceased, presumably other cognitive processes may dictate choice (for example, habitual stimulus-response associations), which would not be appropriately captured by reinforcement learning models. It would be a separate research question to determine the point at which parameters no longer become predictive, requiring a larger dataset to thoroughly assess. We acknowledge that we did not provide sufficient justification for the learning period used for the modeling. In conjunction with the analysis of early sessions outlined above, we have added the following to the text:

“We investigated differences in the acquisition of each task variant by fitting several reinforcement learning (RL) models to early sessions. Our modeling approach closely follows methods outlined in Langdon et al. (2019), in which a much larger dataset (>100 rats per task) was used to develop the RL models applied here. Due to the comparatively small *n* per group in the current study, we limited our model selection to those previously validated in Langdon et al. (2019), with minor extensions. As in previous work, models were fit to valid choices from the first five sessions. As training continues, the impact of outcomes on subsequent choice should decline, and parameter values may evolve over time (e.g., decreasing learning rate). To target the period of learning during which outcomes have maximal influence over choice, and parameters likely have fixed values, we limited our analyses to the first five sessions.”

Concerning the figures, could you consider replacing or including with the bar plots, the full distribution of individual dots, or a violin plot, something to better capture the distribution of the data. This would be particularly beneficial for Figure 2B the risk score which, without a distribution suggests all individuals are optimal, something which in the text claim is not the case.

Individual data points have been added to the relevant figures.

Is this not a case of compositional data where ANOVA is definitely not an appropriate method (compositional data consist in reporting proportions of different elements in a whole, eg this rock is 60% silicate, 20% man-made cement, etc.) because of violation of normality and mostly dependence between measurements (the sum must be 100% as in your case where knowing the proportions of P1, P2 and P3, I automatically deduce P4). I leave to you the care of finding a potential alternative. In any case, I also had difficulties understanding the varying degrees of freedom of the different reported F statistics which worry me that this has not been done properly.

This is a fair criticism, as choice proportions across P1-P4 are not fully independent. While alternative approaches do exist, there is no widely adopted or straightforward method that has been validated for this task. Accordingly, ANOVA remains the standard analytical approach for this task, as it facilitates comparison with previous work and is readily understood by readers. As mentioned in the methods, an arcsine transformation was applied to the proportional data to mitigate issues associated with bounded measures (i.e., summing to 100%). We thank the reviewer for drawing our attention to the discrepancies in the degrees of freedom – these have now been corrected.